# NeuroChains: Extract Local Reasoning Chains of Deep Neural Nets

## Abstract

We study how to explain the main steps/chains of inference that a deep neural net (DNN) relies on to produce predictions in a local region of data space. This problem is related to network pruning and interpretable machine learning but the highlighted differences are: (1) fine-tuning of neurons/filters is forbidden: only exact copies are allowed; (2) we target an extremely high pruning rate, e.g., $\geq 95\%$; (3) the interpretation is for the whole inference process in a local region rather than for individual neurons/filters or on a single sample. In this paper, we introduce an efficient method, NeuroChains, to extract the local inference chains by optimizing a differentiable sparse scoring for the filters and layers to preserve the outputs on given data from a local region. Thereby, NeuroChains can extract an extremely small sub-network composed of filters exactly copied from the original DNN by removing the filters/layers with small scores. We then visualize the sub-network by applying existing interpretation technique to the retained layer/filter/neurons and on any sample from the local region. Its architecture reveals how the inference process stitches and integrates the information layer by layer and filter by filter. We provide detailed and insightful case studies together with three quantitative analyses over thousands of trials to demonstrate the quality, sparsity, fidelity and accuracy of the interpretation within the assigned local regions and over unseen data. In our empirical study, NeuroChains significantly enriches the interpretation and makes the inner mechanism of DNNs more transparent than before.

## 1 Introduction

Deep neural networks (DNNs) greatly reshape a variety of tasks — object classification, semantic segmentation, natural language processing, speech recognition, robotics, etc. Despite its success on a vast majority of clean data, DNNs are also well-known to be sensitive to small amounts of adversarial noises. The lack of sufficient interpretability about their success or failure is one major bottleneck of applying DNNs to important areas such as medical diagnosis, public health, transportation systems, financial analysis, etc.

Interpretable machine learning has attracted growing interest in a variety of areas. The forms of interpretation vary across different methods. For example, attribution methods (Bach et al., 2015; Sundararajan et al., 2017; Shrikumar et al., 2017; Montavon et al., 2017; Kindermans et al., 2017; Smilkov et al., 2017) produce the importance score of each input feature to the output prediction for an given sample, while some other methods (Zeiler & Fergus, 2014; Simonyan et al., 2013; Erhan et al., 2009) aim to explain the general functionality of each neuron/filter or an individual layer regardless of the input sample. Another line of works (Ribeiro et al., 2016; Wu et al., 2018; Hou & Zhou, 2018) explain DNNs in a local region of data space by training a shallow (e.g., linear) and easily interpretable model to approximate the original DNN on some locally similar samples. Thereby, they reduces the problem to explaining the shallow model. These methods essentially reveal the neuron to neuron correlations (e.g., input to output, intermediate layer/neuron to output, etc), but they cannot provide an overview of the whole inference process occurring inside the complicated structure of DNNs.

In this paper, we study a more challenging problem: *Can we unveil the major hidden steps of inference in DNNs and present them in a succinct and human-readable form?* Solving this problem helps to answer many significant questions, e.g., which layer(s)/neuron(s) plays the most/least im-

portant role in the inference process? Do two similar samples really share most inference steps? Do all samples need the same number of neurons/layers to locate the key information leading to their correct predictions? How/when/where does a failure happen during the inference on a DNN? Which neuron(s)/layer(s)/feature(s) are shared by different samples from the same region even when their labels differ? Are DNNs using entirely different parts for data from different local regions? Some of them are related to other problems such as network pruning (Han et al., 2015; Li et al., 2016) and neural architecture search (NAS) (Zoph & Le, 2017). For example, are winning tickets (Frankle & Carbin, 2018; Liu et al., 2018) universal over different local regions and classes? Does the weight sharing scheme in recent NAS methods (Pham et al., 2018; Liu et al., 2019; Ying et al., 2019) limit the searching space or quality?

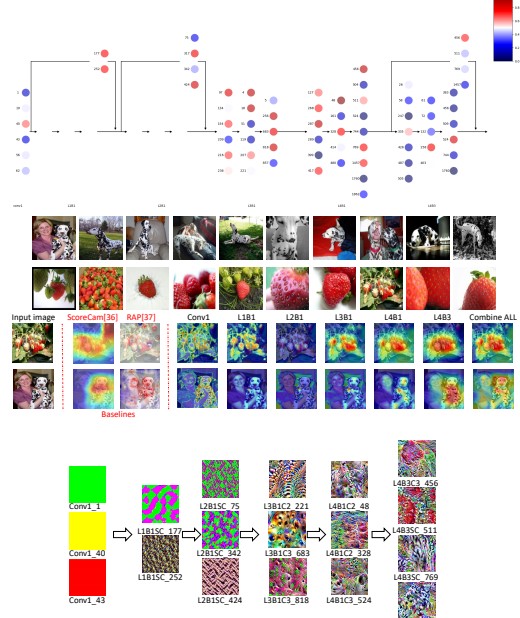

We develop an efficient tool called NeuroChains to extract the underlying inference chains of a DNN for a given local data region. Specifically, we aim to extract a much smaller sub-network composed of a subset of neurons/filters exactly copied from the original DNN and whose output for data from the local region stays consistent with that of the original DNN. In experiments, we assume the data from the same classes reside in a local region. While the selected filters explain the key information captured by the original DNN when applied to data from the local region, the architecture of the sub-network stitches these information sequentially, i.e., step by step and layer by layer, and thus recover the major steps of inference that lead to the final outputs. Despite its combinatorial nature, we parameterize the sub-network as the original DNN with an additional score multiplied to each filter/layer's output featuremap. Thereby, we formulate the above problem of sub-network extraction as optimizing a differentiable sparse scoring of all the filters and layers in order to preserve the outputs on all given samples.

The above problem can be solved by an efficient back-propagation that only updates the scores with fixed filter parameters. The objective is built upon the Kullback–Leibler (KL) divergence between the sub-network's output distribution and that of the original DNN, along with an $\ell_1$ regularization for sparse scores over filters. We further use a sigmoid gate per layer to choose whether removing the entire layer. The gate plays an important role in reducing the sub-network size since most local regions do not rely on all the layers. In practice, we further apply a thresholding to the sparse scores to obtain an even smaller sub-network and employ an additional fine-tuning to the filter scores on the sub-network. We illustrate the sub-network's architecture and visualize its filters and intermediate-layers' featuremaps by existing methods (Mundhenk et al., 2019; Erhan et al., 2009).

Figure 1: Inference chain by NeuroChains for ResNet-50 (pre-trained on ImageNet) when applied to 20 test images of "dalmatian" and "strawberry". **Top:** The sub-network retains only 13/67 layers and 75/26560 filters of the ResNet-50. The scores for selected filters are represented using the colormap on the top-right. **Middle:** The per-layer featuremaps generated by SMOE (Mundhenk et al., 2019) show a clear trends of firstly extracting the local patterns (dots on dalmatian and strawberry) and gradually covering a global shapes of the classes. **Bottom:** Filters with the largest scores are visualized using the method by Erhan et al. (2009). In shallower layers, L4B1C2_48 and L4B1C3_524 capture a more local black spot pattern of the dalmatian, L3B1C3_683 captures the eyes and nose patterns; L3B1C3_818 extracts the local color pattern of strawberry. In the last bottleneck layer, L4B3SC_769 and L4B3SC_1457 capture the global patterns of dalmatian's black and white fur; L4B3SC_456 and L4B3SC_511 captures the main shape and color of strawberry. It shows an inference chain for strawberry: L1B1SC_177 → L2B1SC_75 and L2B1SC_342 → L3B1C3_818 → L4B1C2_328 → L4B3C3_456 and L4B3SC_511.

NeuroChains is a novel pruning technique specifically designed for interpreting the local inference chains of DNNs. As aforementioned, it potentially provides an efficient tool to study other problems in related tasks. However, it has several **fundamental differences to network pruning and exist-**

**ing interpretation tasks**, which deter methods developed for these two problems from addressing our problem. **Comparing to network pruning**: (1) fine-tuning is not allowed in NeuroChains; (2) it targets a much larger pruning rate for succinct visualization, e.g., $\geq 95\%$ for VGG-19 and $\geq 99\%$ for ResNet-50 on ImageNet with $\leq 200$ filters remained; (3) it is for data from a local region instead of the whole data distribution. **Comparing to mainstream interpretation tasks**: (1) NeuroChains produces an interpretation of the entire inference process in a local region (which may contain different classes) rather than of one neuron/filter or output on/around a single sample; (2) the sub-network's architecture provides orthogonal information to the importance of individual neurons/filters.

## 2 RELATED WORKS

**Interpretable machine learning** methods can be mainly categorized into the ones aiming to evaluate the importance of each input feature of a single sample and the ones explaining individual neurons/filters. Approaches in the first category usually rely on certain back-propagation from the DNN's output to derive an importance score for each input feature or hidden node. Earlier works are based on the back-propagated gradients, e.g., deconvolution (Zeiler & Fergus, 2014), back-propagation (Simonyan et al., 2013) and guided back-propagation (Springenberg et al., 2014). Sundararajan et al. (2017) proposed to (approximately) calculate the integral of the gradients along a path between a baseline point and the input sample, which ensures the sensitivity and implementation invariance lacking in some previous methods. More recent methods propose novel back-propagation rules to directly derive the attribution scores of neurons from output to input, e.g., DeepLIFT (Shrikumar et al., 2017), deep Taylor decomposition (Montavon et al., 2017), and layer-wise relevance propagation (LRP) (Bach et al., 2015).

Methods in the second class treat DNNs as black boxes and seek a simple model to explain how the DNN's output changes in a local region. For example, they add perturbations to different parts of the input to evaluate how the perturbations change the output (Ancona et al., 2017), which reflect the importance of different parts. Zeiler & Fergus (2014) covered different parts of the input image with a gray square, which led to different prediction probabilities on the true class. Instead, Zintgraf et al. (2017) replaced each patch in the input image with the surrounding patch and tracked the induced changes in the output. In LIME (Ribeiro et al., 2016), Ribeiro et al. trained a sparse linear model on noisy input-output pairs as a local surrogate approximating the original DNNs, where the sparse weights are used to explain the importance of input features. As mentioned before, our main difference to the above methods are we explain DNNs for a local region of multiple samples and we further explain how DNNs step by step integrate the information of important filters/neurons.

**Network pruning** (Han et al., 2015; Li et al., 2016) remove redundant neurons/nodes or connections/weights from a pre-trained DNN and fine-tune the sub-network. Structural pruning removes whole layers/channels/filters/neurons according to a certain norm of the associated weights (Li et al., 2016) or sparsity (Hu et al., 2016). In contrast, Frankle & Carbin (2018); Liu et al. (2018) prune a DNN during its training. Luo et al. (2017) apply pruning to two adjacent convolution layers at each time to take the dependency between the two layers into account. Several recent works empirically verify "lottery ticket hypothesis", i.e., there exists sub-networks (i.e., winning tickets) that can reach comparable generalization performance as the original DNN if re-trained. In contrast, the sub-network extracted by NeuroChains cannot be fully re-trained since it has to preserve the original DNN's filters, and our goal is to retain the generalization performance only for a local region.

## 3 NEUROCHAINS

### 3.1 PROBLEM: EXTRACT LOCAL INFERENCE CHAINS AS SUB-NETWORKS

Although the DNNs widely used nowadays are usually composed of hundreds of layers and millions to billions of hidden nodes. *When applied to samples from a local region in data space, it is plausible that its inference process mainly relies on a small subset of layers/neurons/filters.* In this paper, we verify this conjecture by developing an efficient and practical algorithm, i.e., NeuroChains, to extract the subset and its underlying architecture as a sub-network whose weights/filters are selected and exactly copied from the original DNN while its outputs in a given local region retain the ones

produced by the original DNN. Although DNNs are usually non-smooth in definition if using a non-smooth piece-wise activation such as ReLU, when trained with the commonly used techniques, e.g., data augmentation, mix-up, dropout, the resulted DNNs are relatively smooth in a sufficiently small local region.

In order to preserve the original inference chains, we do not allow any fine-tuning or re-training on any filter or the the weight vector corresponding to any neuron: they can only be exactly copied from the original DNN. Let $F(\cdot; \{W^\ell\}_{\ell=1:L})$ (a mapping from input to output) denote the original DNN, $W^\ell$ represents the set of filters/weight vectors in layer-$\ell$, and $W^\ell[i]$ represents the $i^{th}$ filter/weight vector in layer-$\ell$. Any sub-network fulfilling our above requirement can be defined and parameterized by an indicator vector $M^\ell$ per layer, whose each entry is a $\{0, 1\}$ value indicating whether retaining the associated filter/neuron in $W^\ell$. We further define operator $\circ$ as

$$(W^\ell \circ M^\ell)[i] \triangleq \left\{ \begin{array}{ll} W^\ell[i], & M^\ell[i] = 1; \\ \mathbf{0}, & M^\ell[i] = 0. \end{array} \right. \tag{1}$$

Thereby, $\{M^\ell\}_{\ell=1:L}$ defines a qualified sub-network for inference chain and its weights are $\{W^\ell \circ M^\ell\}_{\ell=1:L}$, where we extend the operator $\circ$ to make $W \circ M = \{W^\ell \circ M^\ell\}_{\ell=1:L}$ given the original DNN's weights $W = \{W^\ell\}_{\ell=1:L}$. Given a set of samples $\mathcal{X}$ drawn from a local region of data space, we can formulate the problem of finding an inference chain as the following combinatorial optimization, which aims to find the most sparse indicator $M$ (i.e., the sub-network with the fewest filters retained) that does not change the outputs of the original DNN for $\forall x \in \mathcal{X}$, i.e.,

$$\min_{\{M^\ell\}_{\ell=1:L}} \sum_{\ell=1}^{L} \|M^\ell\|_1 \quad \text{s.t. } F(x; W) = F(x; W \circ M), \forall x \in \mathcal{X}. \tag{2}$$

However, directly solving this combinatorial optimization is impractical since the possible choices for $M^\ell$ is of exponential number. In this paper, we relax the 0-1 indicator vector $M^\ell$ to a real-valued score vector $S^\ell$ of the same size. We define an operator $\odot$ applied to $W^\ell$ and its associated scores $S^\ell$ as

$$(W^\ell \odot S^\ell)[i] \triangleq S^\ell[i] \cdot W^\ell[i]. \tag{3}$$

Note we do not limit entries in $S^\ell$ within $[0, 1]$ due to the possible redundancy among filters in the original DNN, i.e., there might be filters of similar functionality for the given samples and a preferred pruning should be able to only preserve one of them and multiply it by the number of those redundant filters in the sub-network. In addition, less constrains are easier to handle in optimization and helpful to find sub-network whose outputs are closer to that of the original DNN, since the class of sub-networks with parameters $W \odot S$ includes all the sub-networks with parameters $W \circ M$. Hence, we relax the challenging combinatorial optimization to the following unconstrained continuous optimization, i.e.,

$$\min_{\{S^\ell\}_{\ell=1:L}} \frac{1}{|\mathcal{X}|} \sum_{x \in \mathcal{X}} l(F(x; W), F(x; W \odot S)) + \lambda \sum_{\ell=1}^{L} \|S^\ell\|_1, \tag{4}$$

where $l(\cdot, \cdot)$ is a loss function aiming to minimize the distance between the original DNN's output $F(x; W)$ and the sub-network's output $F(x; W \odot S)$. In our experiments, for classification, we use KL-divergence between the output distributions over classes, where the two output distributions are computed by applying softmax to $F(x; W)$ and $F(x; W \odot S)$ respectively, i.e.,

$$l(F(x; W), F(x; W \odot S)) = D_{KL}(\text{softmax}(F(x; W))\|\text{softmax}(F(x; W \odot S))). \tag{5}$$

In addition, empirical evidences (Krueger et al., 2017; Singh et al., 2016) show that for most samples there exist some layers that can be entirely removed without changing the final prediction. Hence, only a few hard and confusing samples need more delicate features, while most other samples can be correctly classified based on simple patterns from shallower layers. Therefore, in NeuroChains, we apply a sigmoid function with input score $\alpha^\ell$ as a gate $G^\ell$ determining whether removing the entire layer-$\ell$ during pruning, i.e.,

$$G^\ell = 1/\left[1 + \exp(-\alpha^\ell/T)\right], \tag{6}$$

where $T$ is a temperature parameter. With a gate $G^\ell$ applied after each layer-$\ell$ whose input and output has the same size (which is common in many DNNs), we can recursively define the input

$H^{\ell+1}(\cdot)$ to the next layer-$(\ell + 1)$, i.e., $H^{\ell+1}(x; \{W^{\ell'} \odot S^{\ell'}, \alpha^{\ell'}\}_{\ell'=1:\ell}) =$

$$\begin{cases} G^\ell \cdot F^\ell(H^\ell; W^\ell \odot S^\ell) + (1 - G^\ell) \cdot H^\ell(x; \{W^{\ell'} \odot S^{\ell'}, \alpha^{\ell'}\}_{\ell'=1:\ell-1}), & \text{if input size = output size} \\ F^\ell(H^\ell; W^\ell \odot S^\ell) & \text{otherwise} \end{cases}$$
(7)

where $F^\ell(H^\ell; W^\ell \odot S^\ell)$ denotes the output of layer-$\ell$. The reason to use a gate here is that we expect to either remove the whole layer or retain it without adding an extra shortcut (which will change the original DNN's architecture). We apply a $T < 1$ to sharpen the gate values. Since we prefer to remove non-informative layers, in the objective, we add another regularization $\alpha^\ell$ to encourage the removal of entire layers (because decreasing $\alpha$ reduces $G^\ell$ and thus increase the chance of layer removal). Therefore, the final optimization for NeuroChains is

$$\min_{\{S^\ell, \alpha^\ell\}_{\ell=1:L}} \frac{1}{|\mathcal{X}|} \sum_{x \in \mathcal{X}} l(F(x; W), H^{L+1}(x; \{W^{\ell'} \odot S^{\ell'}, \alpha^{\ell'}\}_{\ell'=1:L})) + \lambda \sum_{\ell=1}^{L} \|S^\ell\|_1 + \lambda_g \sum_{\ell=1}^{L} \alpha^\ell, \quad (8)$$

Our objective above is similar to the one used in Network Slimming (Liu et al., 2017) but we optimize it for a local region (so we can consider to remove layers) and we do not allow fine tuning on weights $W$.

## 3.2 Algorithm

Our algorithm is simply a standard back-propagation for the optimization problem in Eq. (8), which produces sparse scores for filters and gate values for layers. Note the weights in $W$ are fixed and the backpropagation only updates $S$. We initialize the filter scores $S = \mathbf{1}$ so

Table 1: Information of pre-trained DNNs in this paper.

| Statistics | ResNet-50 | VGG-19 |
|---|---|---|
| Top-1 test accuracy | 76.5% | 72.9% |
| Test images/sub-networks | 10000/1688 | 10000/1746 |
| Convolutional filters | 26560 | 4480 |
| Parameters of Conv-layers | 23454912 | 20018880 |
| Parameters of FC-layers | 2048000 | 123633664 |

$W \odot S = W$ at the beginning of optimization. We initialize the gate score $\alpha^\ell = 0$ for all $\ell = 1 : L$ so $G^\ell = 0.5$ at the beginning, i.e., the probabilities to remove or to retain a layer is equal. For classification, we set loss $l(\cdot, \cdot)$ to be the KL-divergence between the output distributions of the original DNN and the sub-network. After convergence of the optimization, we then apply a simple thresholding to these scores to further remove more filters and layers: (1) we remove the filters with score under a threshold $\tau$; (2) we remove layer-$\ell$ if $G^\ell < 0.5$. This yields a sufficiently small sub-network architecture but the scores might be sub-optimal in preserving the original DNN's outputs within the local region. Therefore, we further fine-tune the nonzero scores in $S$ (note filters are always fixed), i.e, the scores for the retained filters/layers, by minimizing Eq. (8) without the two regularizations, if the first term in Eq. (8) exceeds a threshold $t$. Given a sub-network produced by NeuroChains, we then visualize its architecture and scores as the structure of the inference chains. Moreover, we visualize the retained filters and layers on the chains by their activation patterns and featuremaps, respectively, using existing interpretation methods (Mundhenk et al., 2019; Erhan et al., 2009).

## 4 Experiments

In experiments, we apply NeuroChains to extract the inference chains of two widely-adopted CNNs pre-trained on ImageNet, i.e., VGG-19 and ResNet-50. We provide the basic information of the two DNNs in Table 1. In the following, We will present **three quantitative analyses over hundreds of case studies**, which show that (1) NeuroChains is capable to produce sub-networks retaining only $< 5\%$ of filters and meanwhile preserve the outputs of the original DNN in most cases; (2) every filter selected by NeuroChains is important to preserving the outputs since removing one will leads to considerable drop in performance; (3) the sub-network extracted based on a finite number of samples can be generalized to unseen samples in nearby regions. To demonstrate the effectiveness of NeuroChains in local regions in the non-smooth raw-input space, we evaluate the sub-network on the adversarial examples of each sample. We also study the firing behaviour of filters in the original network and the sub-network. We will then provide several detailed and insightful case studies and visualizations of extracted sub-networks for different local regions.

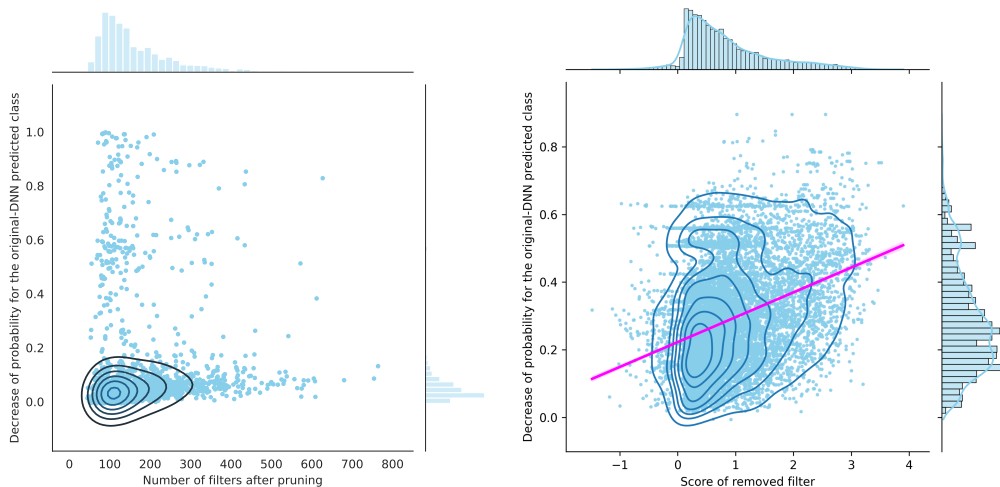

Figure 2: **Left: Size and fidelity (how well the sub-networks preserve the original DNN's outputs)** of 1500 sub-networks extracted by NeuroChains for ResNet-50 in different case studies (local regions). The x-axis refers to the number of retained filters, while the y-axis is the induced decrease of probability on the original-DNN predicted class. It shows that NeuroChains can usually find very small sub-networks and meanwhile preserve the original DNN's outputs. **Right: Faithfulness (filter score and degeneration by removing the filter)** of 783 sub-networks, each extracted by NeuroChains on 20 uniform samples randomly drawn from two classes. The x-axis refers to the filter scores, while the y-axis denotes the decrease of the sub-network's probability on the original-DNN predicted class after removing a filter from the sub-network. It shows that the sub-networks suffer from more degeneration if removing a filter with higher score (magenta line and the shaded areas show strong linear correlation between the two). Hence, the scores faithfully reflect the importance of filters in explaining the original DNNs.

## 4.1 IMPLEMENTATION DETAILS

We implement NeuroChains by PyTorch (Paszke et al., 2017). In every case study, we firstly randomly sample 2 classes and then randomly sample 10 images from each class's images that the original DNN has high confidence, e.g., $\geq 95\%$. Note high confidence does not guarantee the correctness since it can be associated with a wrong class. We choose images with high confidence since the inference on them mostly represent the original DNN's inference chains in the region. We apply inference on those 20 images and their outputs are used in solving the optimization of Eq (8) in order to extract the local inference chain in the form of a sub-network. For models with shortcuts, e.g., ResNet-50, the sigmoid gate is applied to prune a bottleneck block rather than a layer. A layer inside a block will be removed if the scores of all filters in the layer are nearly 0.

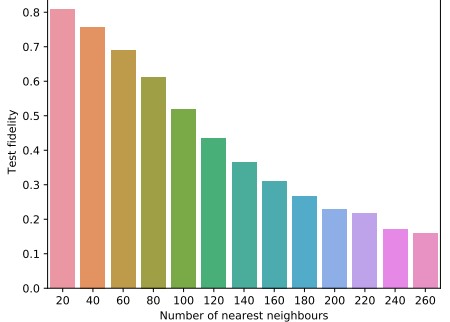

Figure 3: **Averaged test fidelity (stability)** of 1500 sub-networks extracted by NeuroChains for VGG-19 when applied to unseen images in nearby local regions. For each sub-network and every image used to extract it, we rank images from the validation set of Deng et al. (2009) by their distance to the image (in VGG-19 penultimate-layer's output space) and obtain its K-nearest neighbors. We evaluate the test fidelity (accuracy of preserving the original DNN's predicted class) of the sub-network over these unseen K-NN images for all the images used to extract it (duplicates removed). The x-axis refers to 20K since each sub-network is extracted for 20 samples, while the y-axis is the test fidelity averaged over all the 1500 sub-networks.

We use Adam optimizer for both the optimization of Eq (8) and fine-tune phases for filter/layer scores. We use a fixed learning rate 0.005 in the former and 0.01 in the latter. We set temperature $T = 0.2$ in the sigmoid gate (Eq. (6)) to encourage the gate value $G^\ell$ close to either 0 or 1, and the threshold $\tau$ to goal scores is set to 0.1 so that the outputs of sub-networks are as consistent as possible. We only tried a limited number of choices on tens of experiments, and chose the best combination balancing the fidelity and sub-network size, and then applied it to all other experiments without further tuning. In particular, we tried $\tau \in \{0.01, 0.1, 0.5\}$, $\lambda \in \{0.001, 0.005, 0.01, 0.1\}$, and $\lambda_g \in \{1, 2, 5\}$. For different models, the weights of two penalties in Eq. (8) are different. For VGG-19, we use $\lambda = 0.007$ and $\lambda_g = 2$. While we choose $\lambda = 0.0065$ and $\lambda_g = 0.9$ for ResNet-50.

This choice performs consistently well and robust on all other experiments. We fine-tune $S$ in the pruned network if the KL-divergence after training is greater than threshold $t = 0.1$. The iteration steps of training and fine-tune is 300 and 50 respectively. We stop training or fine-tuning when the loss difference is less than 0.02. It costs only $\sim$ 90s for VGG-19 and $\sim$ 55s for ResNet50 to extract a sub-network on a single GPU since we only optimize a few number of scores.

## 4.2 QUANTITATIVE ANALYSES

In Melis & Jaakkola (2018), they propose three criteria to evaluate the interpretation methods for DNNs, i.e., faithfulness, stability and explicitness. In this paper, we extend some of their notations and present three quantitative analyses of NeuroChains over 1500 case studies for different local regions, i.e., (1) **Fidelity**: does the sub-network preserve the original DNN's outputs on the given samples? how does it change for sub-networks of different sizes? (2) **Faithfulness**: how much degeneration on the fidelity will be caused if removing one filter from the sub-network? (3) **Stability**: what is the fidelity (the accuracy of preserving the original DNN's prediction) of the sub-network on unseen test images from nearby regions? In this paper, we evaluate the fidelity and faithfulness by the decreasing amount of probability on the original DNN's predicted class when using the sub-network for inference; we evaluate the stability by evaluating the test fidelity of the sub-network over an image's K-nearest neighbor images from an validation set (details below Figure 14). All the above metrics are averaged over 1500 sub-networks and across all the images used to extract each sub-network.

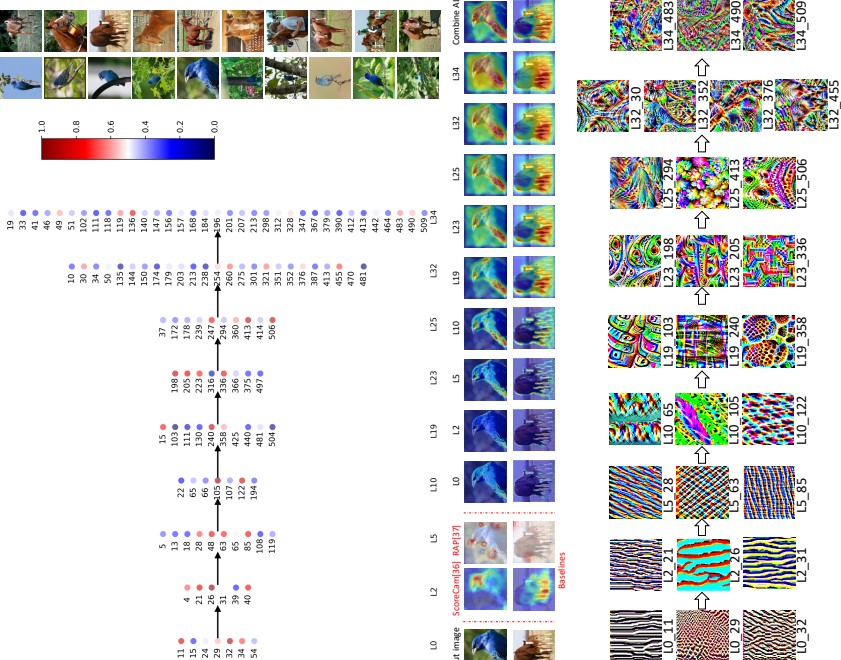

Figure 4: Inference chain by NeuroChains for VGG-19 when applied to images of "Indigo bunting" and "sorrel". **Left:** The sub-network retains only 9/16 layers and 118/4480 filters of the VGG-19. **Middle:** In the SMOE featuremaps, the eyes of both indigo bunting and sorrel, the feathers of indigo bunting, and the legs of sorrel are gradually located as the key features. **Right:** In shallow layers, filters extract local patterns like eyes (L23_198, L23_205 and L25_506) and feathers (L25_294 and L25_413). In the last two layers, L32_30 captures the eyes of sorrel (big and with eyelids) while L32_352 shows the eyes and the whole head of the bird; L32_455 captures the pattern of sorrel legs; L34_490 captures the contour of sorrel's main body. It implies an inference chain for Indigo bunting: L10_105 → L19_103 → L23_205 → L25_506 and L25_294 → L32_352 → L34_483.

The left plot of Figure 2 shows the fidelity for sub-networks of different sizes (measured by the number of filters) that are extracted by NeuroChains. Note most sub-networks only retain $\leq 1\%$ of ResNet-50 for succinct visualization but they preserve the outputs of ResNet-50 with high fidelity.

The right plot in Figure 2 reports the faithfulness of extracted sub-networks, i.e., how a sub-network's performance in preserving the original DNN's output degrade if removing one filter from it, and what is the relationship between this degeneration and the score of the removed filter. The

statistics on 1500 sub-networks in Figure 2(right) show that removing even only one filter from the sub-network can significantly degrade the explanation performance. Hence, NeuroChains usually find the smallest sub-networks without redundancy among retained filters/layers, i.e., every critical inference step is retained. Moreover, the degeneration and the score are strongly and positively correlated, indicating that our optimized scores faithfully reflect the importance of filters in explaining the original DNN. We also present another faithfulness study based on a quotient metric defined below. Let $p, q \in \Delta^c$ ($\Delta^c$ is the probability simplex for $c$ classes) be the output probability vectors of the original neural net and the extracted sub-network respectively for a same input. We define a quotient metric to measure the change of class prediction between $p$ and $q$, i.e.,

$$Q(p, q) = \frac{q[y] - \max_{z \in [c], z \neq y} q[z]}{p[y] - \max_{z \in [c], z \neq y} p[z]}, \quad y \in \arg\max_{z \in [c]} p[z], \tag{9}$$

where $y$ is the predicted class by the original neural net, and $Q(p, q)$ is the quotient of two probability differences computed respectively on the original neural net and the sub-network. In particular, it computes the difference of probabilities for class $y$ and the highest-rated other class. The sign of $Q(p, q)$ indicates whether the predicted class changes (e.g., it changes if $Q(p, q) < 0$) while the magnitude of $Q(p, q)$ measures the change in prediction confidence. The result is given in Figure 8 of Appendix.

We report the average stability of 1500 sub-networks extracted by NeuroChains in Figure 14 of Appendix. It shows that each sub-network can still achieve a high fidelity (the accuracy of preserving the original DNN's output) within a relatively large local region (e.g., 100 unseen samples) around the 20 samples used to extract the sub-network. It therefore indicates that the inference chains extracted for a finite number of samples by NeuroChains can be applied and generalized to other unseen samples from the same region or even the nearby regions. So NeuroChains is an effective solution to the problem raised in the introduction.

In order to evaluate NeuroChains on the local regions in the raw input space, we extract sub-networks for uniformly random drawn samples and then evaluate the sub-networks on these samples' adversarial examples generated by two types of attacks: fast gradient sign method (FGSM) and projected gradient descent (PGD). Figure 5 compares the robustness of the original neural net (LEFT plot) and the extracted sub-networks (RIGHT plot) under different attacks: each of the plots show the histogram of the output probability for the ground-truth class on those samples (original and adversarial). The left plot shows that the two types of adversarial attack are very effective on the original neural net in reducing the probability of ground truth class. In contrast, the right plot shows that the NeuroChains extracted sub-networks are much more robust to the attacks, because the optimization in NeuroChains not only removes the irrelevant filters but also strengthens the important filters by assigning them weights $> 1$. This demonstrates the effectiveness of NeuroChains when applied to local regions in the non-smooth raw-input space, and the extracted sub-networks in this case significantly improves the robustness of the original model in defending adversarial attacks.

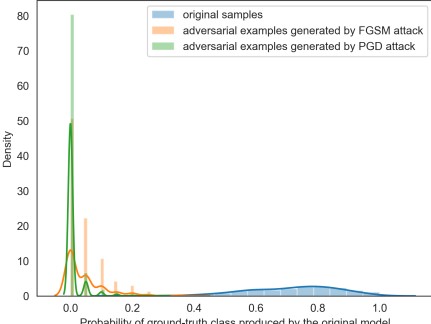 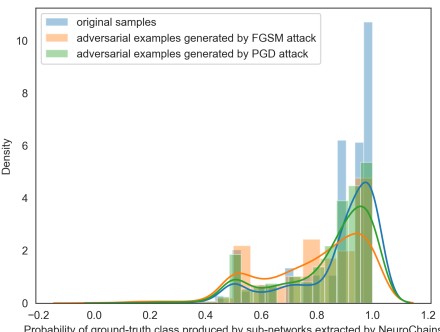

Figure 5: Robustness of original model (LEFT) vs. extracted sub-networks (RIGHT) under different attacks. The firing behaviour of the same filter in the original network and the sub-network is usually different since NeuroChains combines similar filters by assigning scores/weights $> 1$ to some filters. For this reason, one filter's featuremap produced in the sub-network is usually a combination of the featuremaps produced by multiple (pruned) filters from the original network, so the correlation between the same filter's featuremaps in the two networks can be small. In this way, we can better compress the networks to sub-networks with much fewer filters that are easier to interpret while preserving the original network's inference process. To see this phenomenon, we present two case studies in

Figure 6. It shows that each featuremap of the original network's filter only represents one part of the object, while the featuremap of the preserved filter in the sub-network combines all the parts to form a complete representation of the object in the image. More complete quantitative analyses for both VGG-19 and ResNet-50 can be found in the Appendix.

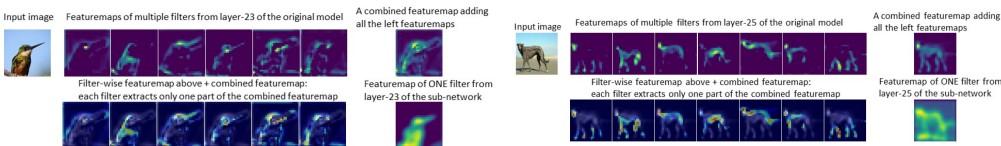

Figure 6: Firing behaviour (featuremaps) of filters in the original network and the sub-network.

## 4.3 CASE STUDIES

We present three case studies of the sub-networks extracted by NeuroChains . For Figure 1 and Figure 4, the data points are from classes which are easy to tell apart while in Figure 7 images are always mis-classified. More case studies are given in Appendix. The visualization in each case study is composed of five parts: (1) the sub-network's architecture and filter scores; (2) the 20 images from 2 classes used to extract the sub-network. The sub-network is supposed to explain the inference chains on images from the same local region of those 20 images; (3) one image from each class and the visualization of the image's per-layer featuremaps in the sub-network, which is produced by SMOE (Mundhenk et al., 2019); (4) for comparison, we also present the interpretations produced by two recent works (Wang et al., 2019; Nam et al., 2019) on the two images; (5) visualization of each filter retained in the sub-network using Erhan et al. (2009) after zoomed-in. In the caption below each case study, we highlight some filters/featuremaps and their related class. We also provide examples of inference chains composed of filters stitched by the sub-network's architecture. They show that NeuroChains considerably enrich the explanation details of DNN's inference process in a local data region. By connecting the important filters from different layers, the extracted sub-network highlights the main steps leading to the output prediction.

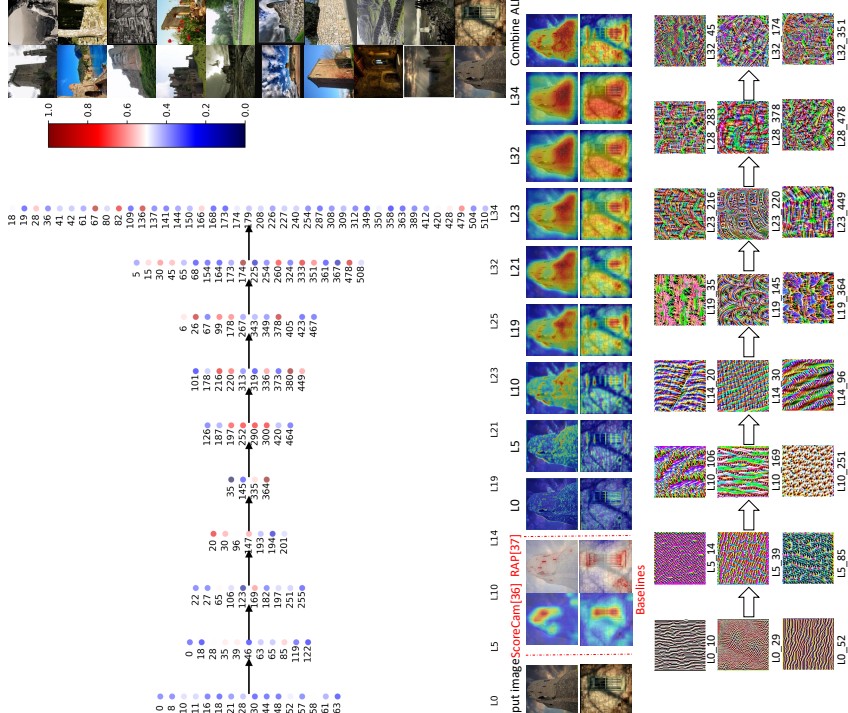

Figure 7: Inference chain by NeuroChains for VGG-19 when applied to images of "Castle" and "Stone Wall" which confuse DNN models. **Left:** The sub-network retains only 10/16 layers and 138/4480 filters of the VGG-19. **Middle:** In the SMOE featuremaps, the wall of castle and the windows of the wall are gradually located as the wrong key features. **Right:** For castle: L14_96 → L19_145 → L23_216 and L23_220 → L28_283 → L32_351; For stone wall: L14_20 → L19_364 → L23_449 → L28_378 → L32_45. Filters like L19_364, L23_216 and L28_378 confuse the model and cause wrong predicts since they capture the patterns of both classes.

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

## 5 APPENDIX

### 5.1 QUANTITATIVE ANALYSIS

We performed 783 experiments each using 20 samples uniformly drawn from two classes (not only the high-confidence samples) and achieved 783 new sub-networks by NeuroChains on VGG-19. We evaluated these newly generated sub-networks using the quotient metric "A quotient of "diff to highest scoring other class (extracted)" / "diff to highest scoring other class (original)" Eq. (9). We visualized the result in Figure 8: The left plot is the histogram of the quotient computed over all the 783×20 samples. The histogram shows that most samples keep the original predicted label after pruning, i.e., NeuroChains can preserve the original DNN's outputs in most cases. Moreover, the number of filters preserved in these sub-networks is $157(mean) \pm 43(std)$, which is small enough to explain. The right plot reports the Faithfulness of NeuroChains in terms of the quotient's sign. We remove each filter from each sub-network and report how many samples' predicted labels are changed after the removal, i.e., the quotient is negative. Each point in the scatter plot corresponds to a sub-network, the x-axis is the score of the removed filter given by NeuroChains, and the y-axis is the proportion of samples with negative quotients. The plot shows a strong linear correlation between the score of the removed filter and the degradation of faithfulness. Since removing filters with high scores results in more samples with predicted class changing after pruning, the score given by NeuroChains measures the importance of filters in DNN inference.

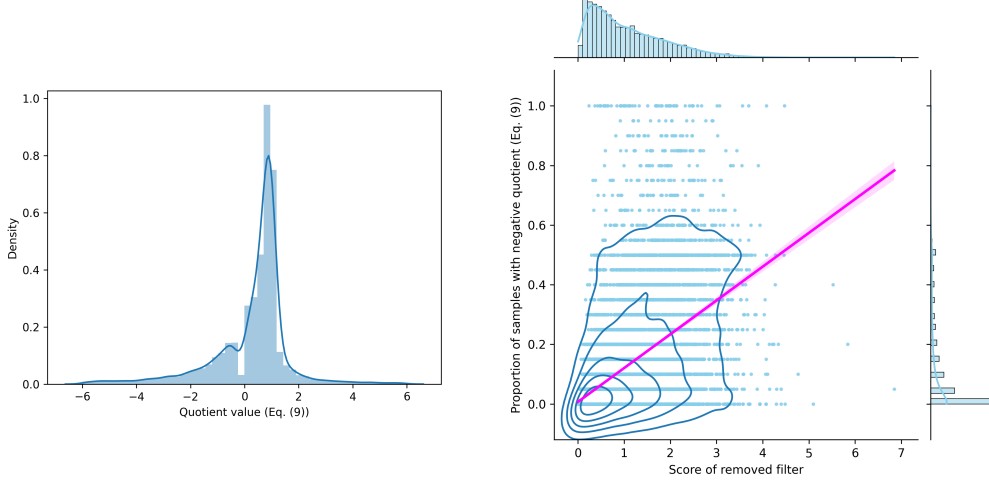

Figure 8: Histogram of the quotient metric in Eq. (9) computed over all the 783×20 samples (LEFT). Faithfulness of NeuroChains in terms of the quotient's sign (RIGHT).

We evaluate the stability of NeuroChains using the nearest neighbours from the penultimate-layer representation space. Because ReLU pattern does not provide an ideal metric to measure the distance of samples, even in the raw input space: (1) the number of ReLU linearity zones grows exponentially with the number of hidden nodes. Most ReLU linearity zones are empty and do not contain any real sample; (2) For the few ReLU linearity zones that do contain samples, each only contains one sample and by large chance its neighboring linearity zones are empty, and this is true for most practical cases as empirical studies suggested. So it is almost impossible to find two samples sharing the same ReLU linearity zone or even close in their ReLU patterns of the first layer; (3) For two ReLU linearity zones that are only different in one facet of their polyhedra (i.e., only one digit of their ReLU patterns flips), their corresponding linear models can still be very different (the linear model is an extreme case of sub-network). Therefore, we speculate that samples close to each other in terms of their ReLU patterns do not share a sufficiently small sub-network preserving their original predictions.

That being said, we evaluated each NeuroChains extracted sub-network of VGG-19 using 20 samples randomly drawn from two classes on the K-th nearest neighbour (NN) of each sample by sorting the Hamming distance on their ReLU patterns. The K-NN samples' prediction cannot be well pre-

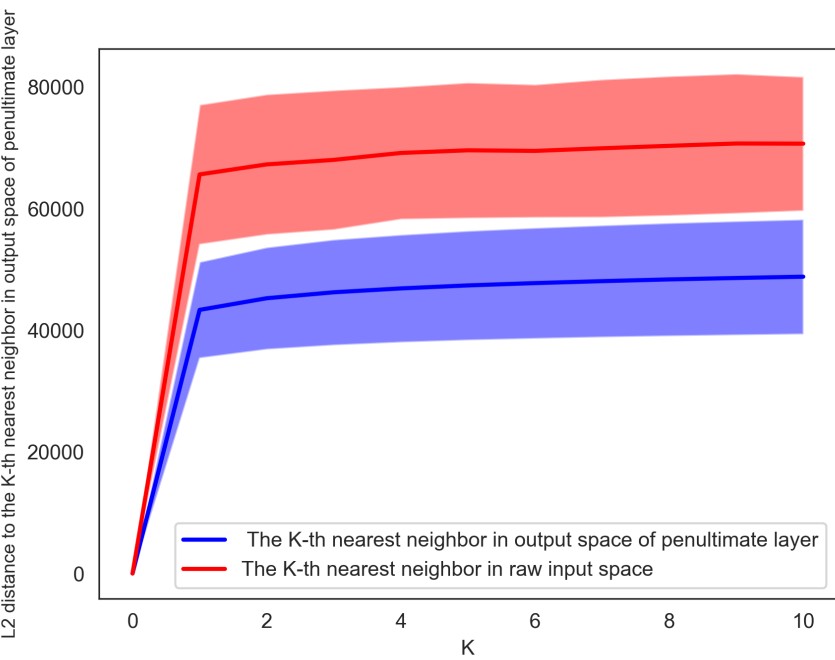

Figure 9: Mean±std of L2 distance in the the penultimate-layer representation space between each sample and its K-nearest neighbours from the penultimate-layer representation space (blue) and ReLU pattern space (red).

served on the sub-networks, because the nearest neighbors in terms of ReLU patterns have very different semantic concepts or classes from the samples that the sub-networks are extracted for. Hence, the local region of ReLU patterns is not a local region on the smooth data manifold. To see this, in Figure 9, for each sample, we computed its L2 distance to the ReLU pattern K-NN sample's penultimate-layer representation for $K = 1, 2, \cdots, 10$ (the red curve reports mean±std), and we compared them with the L2 distance to the K-NN in the penultimate-layer representation space (the blue curve reports mean±std). It shows that the ReLU pattern K-NN has a much larger L2 distance in the semantic space (i.e., penultimate-layer representation), so it is very different in concepts to the original sample. Moreover, we show some examples of the ReLU pattern K-NN images and the penultimate-layer K-NN images for the sample in Figure 10, which show that ReLU pattern K-NN images are much less related to the original sample.

In Figure 11, we show two case studies of comparing SMOE generated heatmaps for the original network and the NeuroChains extracted sub-network. We can see that the patterns extracted by the two networks are consistent and are all critical patterns for the class, e.g., the eyes and fists of kangaroos and the feet and face of the horse. However, compared with the original network, these patterns are strengthened in much shallower layers of the sub-network, producing better interpretations. This observation is also consistent with the result of analysis on adversarial attacks in Figure 5.

In Figure 12, we compare the capability of preserving the original neural network's outputs between NeuroChains and magnitude-based pruning (removing the filters whose output featuremaps' average magnitude (L2 norm) over all considered samples is small). In particular, under the same setting of each experiment in the paper, we prune the original VGG-19 and retain the filters with the largest featuremap magnitude in each layer, 180 in total (more than $157(mean) \pm 43(std)$ filters for sub-networks extracted by NeuroChains), and we then fine-tune the filters' scores/weights as we did for NeuroChains. Figure 12 shows the histogram of the KL divergence between the original output class distribution and the one produced by the sub-networks. For sub-networks generated by NeuroChain, the KL-divergence in most cases stays close to 0, while the output preserving capability of simple pruning is much worse.

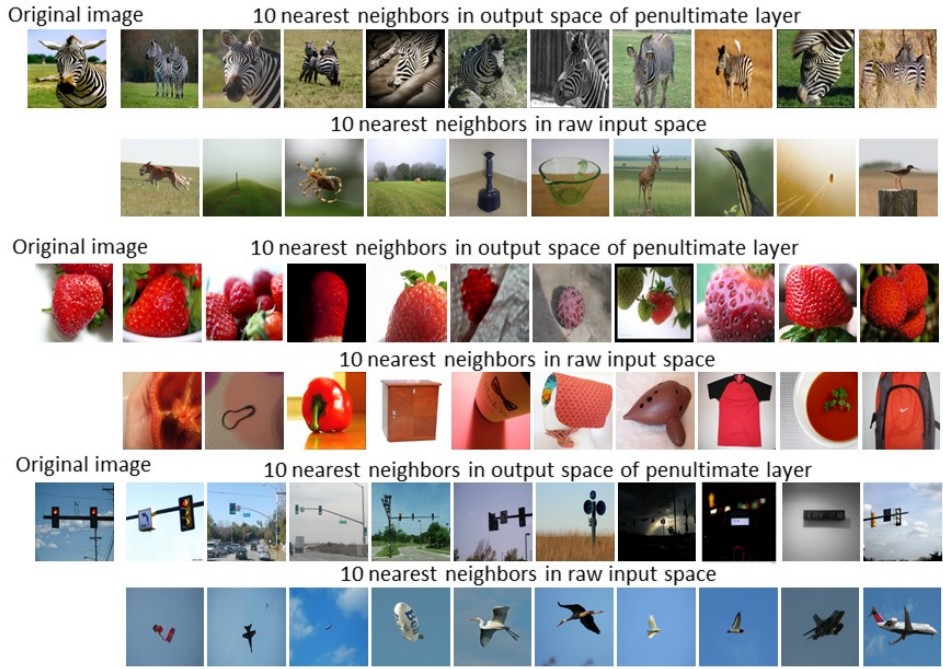

Figure 10: Case studies of an image, its 10-nearest neighbours in the output space of penultimate layer (Top), and its 10-nearest neighbours in the raw input space in terms of Hamming distance between first-layer ReLU patterns (Bottom).

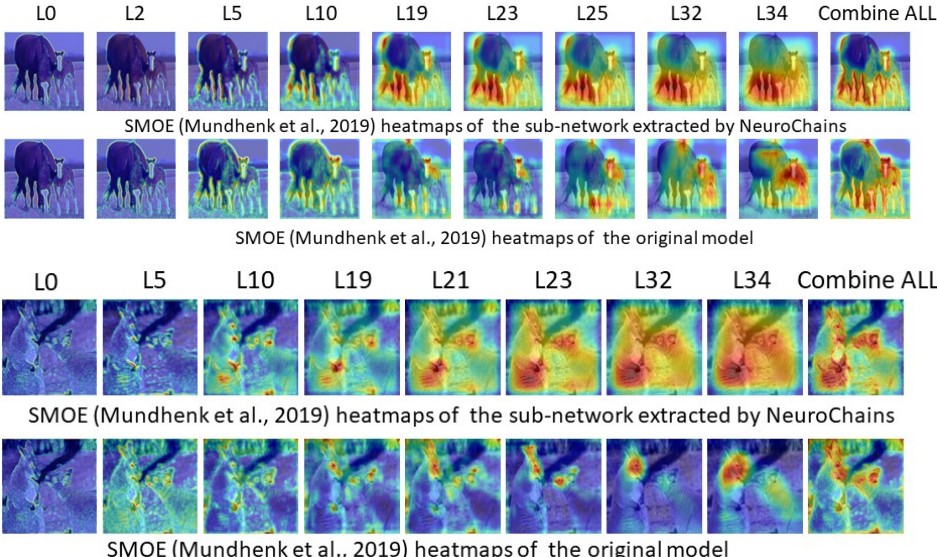

Figure 11: Case studies of SMOE generated heatmaps for the original network and the NeuroChains extracted sub-network.

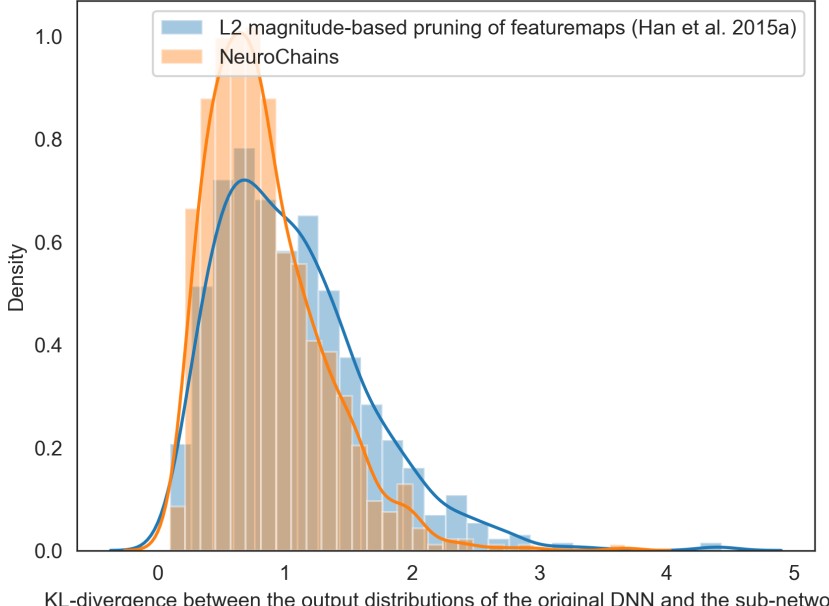

Figure 12: Comparison of NeuroChains and magnitude-based pruning on the capability of preserving the original network's output distribution (smaller KL divergence means better preservation) over $783 \times 20$ uniform samples.

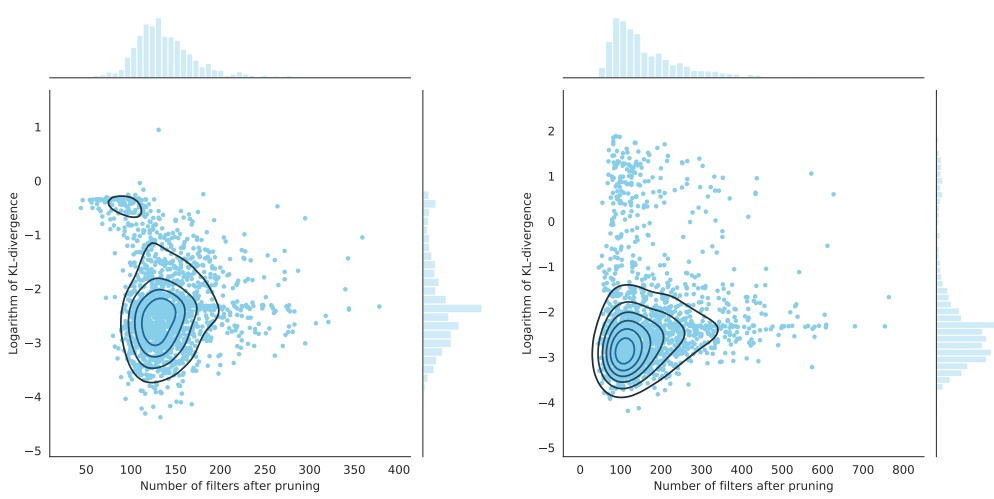

Figure 13: Statistics of the output discrepancy between sub-networks extracted by NeuroChains and the original network: VGG-19 (Left) and ResNet-50 (Right). The x-axis refers to the number of filters in sub-DNNS, the y-axis is the logrithm of KL-divergence between the output distributions produced by the sub-networks and the original network. The KL-divergence for most samples are small, indicating the sub-networks preserve the original network's output distribution for most samples.

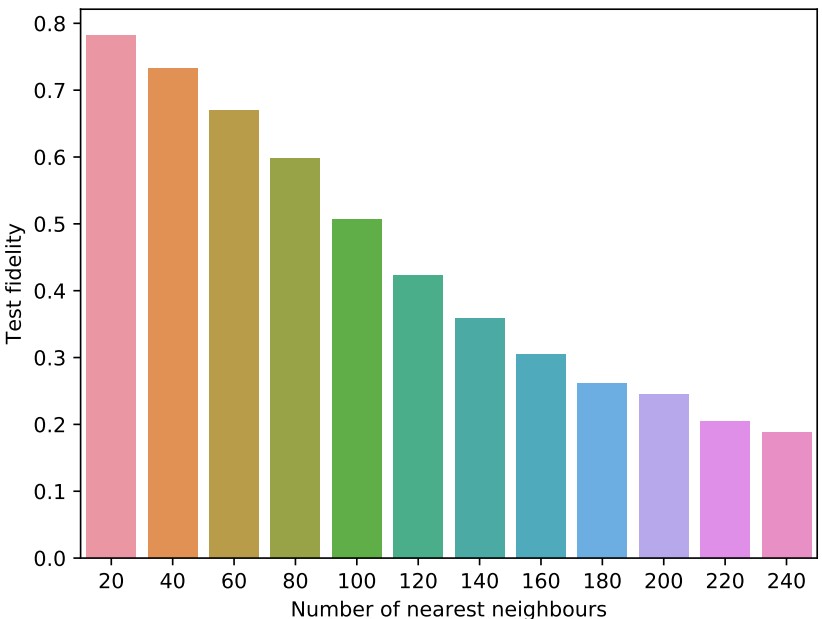

Figure 14: Histogram of stability of sub-networks extracted by NeuroChains for ResNet-50. The x-axis refers to 20K for the K-nearest neighbours of the 20 samples used to extract the sub-network) in the penultimate-layer representation space, while the y-axis is the test fidelity (averaged over all sub-networks), i.e., the accuracy of sub-networks in preserving the predicted class by the original network on the unseem K-nearest neighbours.

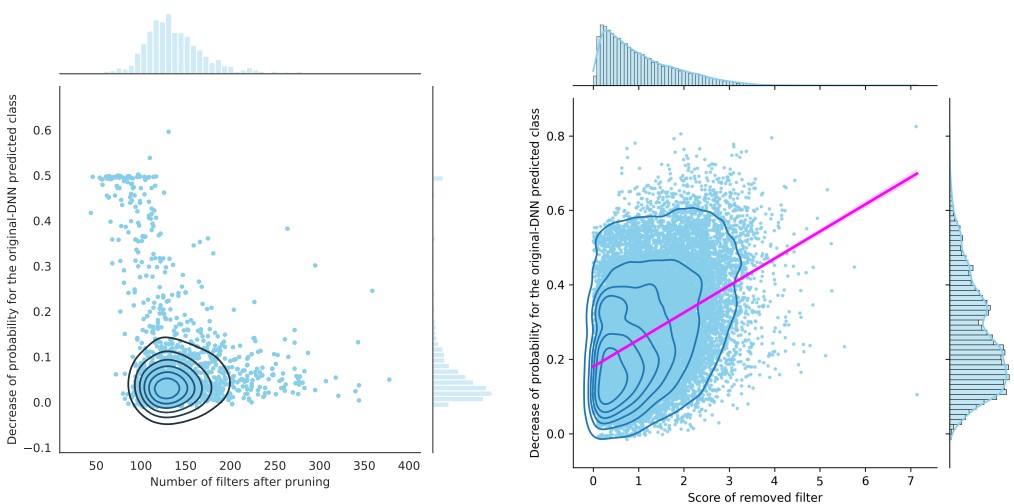

Figure 15: **Left:** Scatterplot with a jointly density estimate of the performance of sub-networks extracted by NeuroChains for VGG-19. Each point corresponds to a sample. The x-axis refers to the number of filters in the sub-network, the y-axis measures the decrease of probability on the original network predicted class. For VGG-19, most sub-networks' output probabilities drop very little regardless of how many filters are retained. **Right:** Scatterplot with a jointly density estimate of faithfulness of sub-DNNs extracted by NeuroChains for VGG-19. The x-axis refers to the scaling score of removed filter, the y-axis is the decrease of average probability for the original-DNN predicted class compared with the complete sub-DNNs. For VGG-19, it seems the higher the score, the more the probability drops. The slope of the magenta line is the linear (Pearson) correlation, while the shaded area around the line represents the confidence interval.

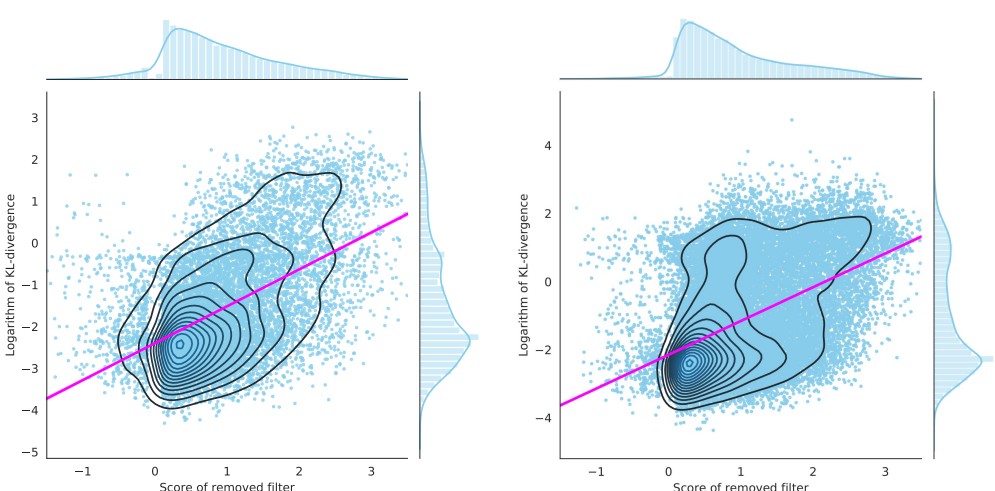

Figure 16: Scatterplot with a jointly density estimate of faithfulness of sub-networks extracted by NeuroChains for VGG-19 (Left) and ResNet50 (Right). The x-axis refers to the scaling score (weight) of removed filter in the sub-networks, the y-axis is the logrithm of KL-divergence between the outputs of the new sub-networks (after removal) and the original sub-networks (before removal). For both VGG-19 and Resnet-50, it shows that the higher the score, the higher the KL-divergence.

## 5.2 MORE DETAILS ABOUT CASE STUDIES

On the sub-network's architecture, we use "L0" to denote the corresponding convolution layer in VGG-19 and "L0_1" to denote the first filter from this layer. For ResNet-50, we further use "L1B1" to denote the first sub-block in the first bottleneck block, "SC" for the shortcut connection and "C1" for the first convolution layer in the sub-block. The redder the node in the sub-network, the larger the scaling score, conversely, the bluer the node, the lower the score. More case studies can be found in the Appendix. In SMOE, Mundhenk et al. propose to measure information at the end of every feature scale and then combined them into a saliency map. We apply this technique to each layer of the sub-network since each layer may prefer different features. In each of our case study, a featuremap-overlaid input image is shown for each layer and for the whole sub-network which is marked as "Combine All". The visualization of each selected filter is achieved by maximizing its activation w.r.t. the input. Afterward, we shows the patterns that the filter aims to detect which is independent of the input image.

## 5.3 CASE STUDIES

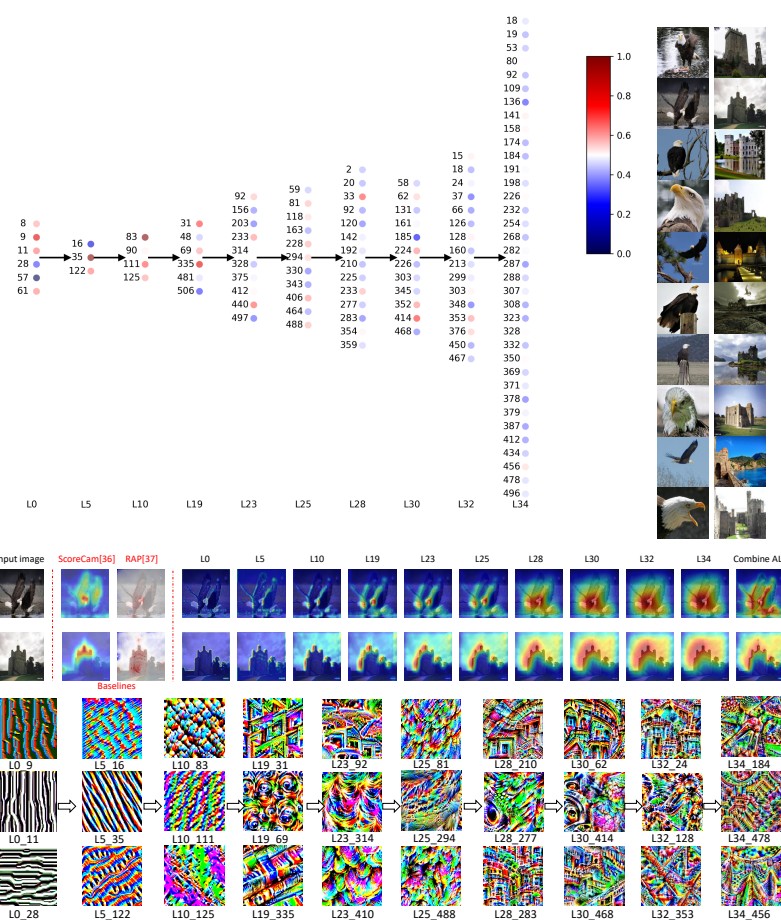

Figure 17: Inference chain by NeuroChains for VGG-19 when applied to images of "bald eagle" and "castle". **Top:** The sub-network retains only 10/16 layers and 118/4480 filters. **Middle:** The per-layer featuremaps generated by SMOE. Since there is nothing similar between bald eagles and castles, it's easy for VGG-19 to tell them apart. Different types of feathers are an important feature of eagle and the contour of the castle is highlighted. **Bottom:** Filters with the largest scores. In shallower layers, L23_314 and L19_69 capture the patterns of feathers and eyes of eagle, which are different from other species. L28_277 in the deeper layer combines the above two patterns. L25_81 identify the half circle of feathers around neck to be key pattern of eagle. L32_24 and L34_184 can be explained as detectors of the whole head and neck of eagle that combines all the patterns detected in previous layers. L23_92 shows the pattern of small room with windows. L28_210, L28_283, L30_62 and L32_24 extract clear patterns of castle. It shows an inference chain for eagle: L10_83 → L19_69, L23_314 and L25_294 → L28_277 → L30_414 → L32_128 → L34_184. It shows an inference chain for castle: L10_125 → L19_31 → L23_92 → L28_210 and L28_283 → L30_62 and L30_468 → L32_24 → L34_478.

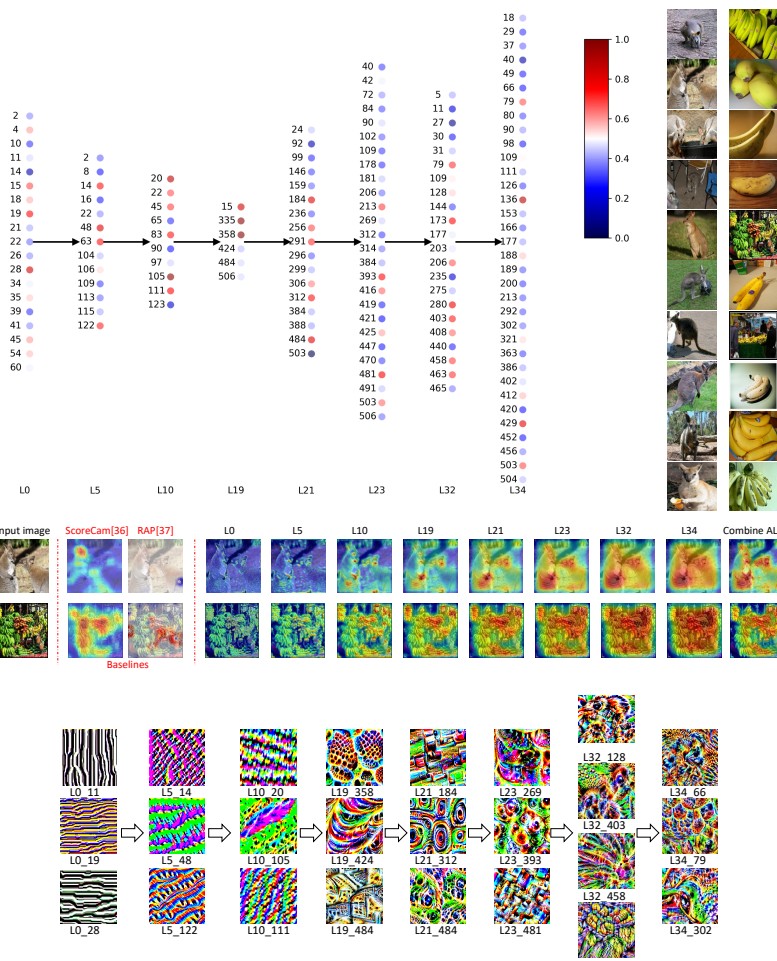

Figure 18: Inference chain by NeuroChains for VGG-19 when applied to images of "kangaroo" and "banana". **Top:** The sub-network retains only 8/16 layers and 148/4480 filters. **Middle:** The per-layer featuremaps generated by SMOE. The black ball-shape pattern exists both in kangaroos and bananas. The hands and eyes of kangaroos are highlighted, and the ends of bananas are lit up. These parts are all black and round in images. **Bottom:** Filters with the largest scores. L21_312, L23_269, L23_393 are all related to the black round pattern. L21_312 shows the basic black round pattern. L23_269 looks like the eyes and noses of animals while in L23_393 these nodes are closely arranged like a hand of bananas. To better distinguish this two class, VGG-19 introduces some key patterns for each class. L32_465 and L34_79 depict the whole image of hands of bananas. L34_66 combine the previous features and show the pattern of animal faces. It shows an inference chain for kangaroo: L10_105 → L19_358 → L21_312 and L21_484 → L23_269 → L32_403 → L34_66.

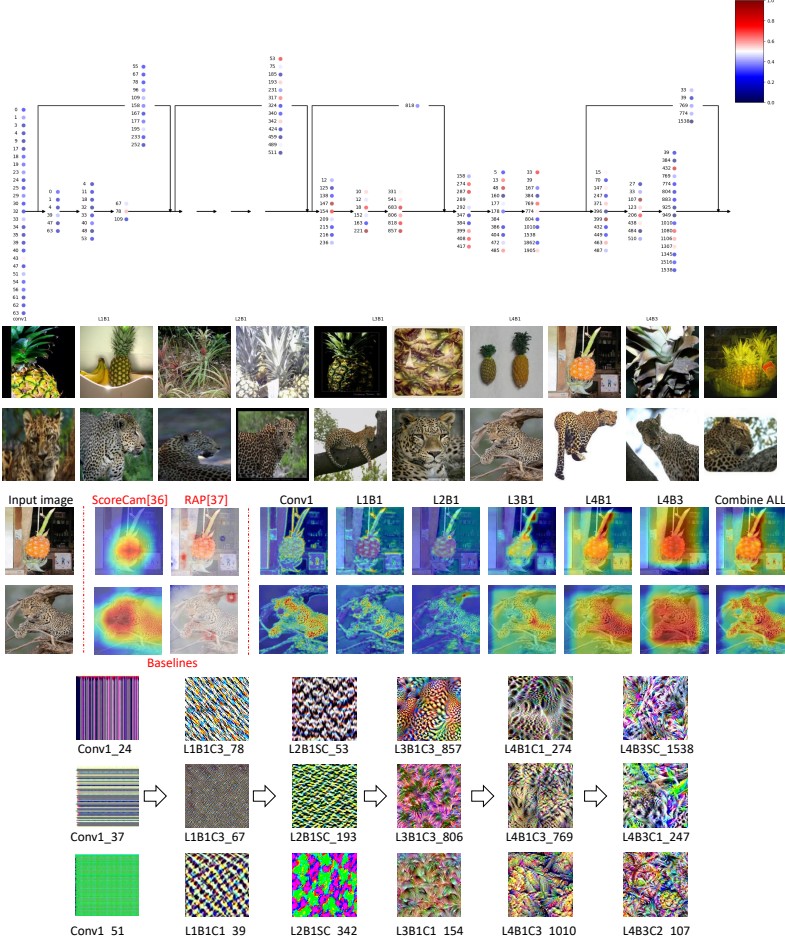

Figure 19: Inference chain by NeuroChains for Resnet-50 when applied to images of "pineapple" and "leopard". **Top:** The sub-network retains only 17/67 layers and 157/26560 filters. **Middle:** The per-layer featuremaps generated by SMOE. Both the body and leaves of the pineapple are highlighted. The special skin texture is enough for ResNet-50 to identify leopard. **Bottom:** Filters with the largest scores. By observing the patterns in the activation maximization result and the highlighted regions in the featuremap, we can find that some filters extract different local patterns appearing at different parts of pineapple. For example, L4B1C3_1010 capture the texture and the color of the main body, L3B1C1_154 capture the patterns of the leaf part. It is interesting to see that L4B3C2_107 is the accurate descriptor of the main body and the leaf parts and thus provide nearly orthogonal features. For leopard, the skin marked with black spots is its most obvious feature. L4B1C1_274 extracts the basic texture and color while L4B3SC_1538 and L4B3C1_247 really show the skin pattern of the leopard. It shows an inference chain for pineapple: L2B1SC_342 → L3B1C3_806 and L3B1C1_154 → L4B1C3_1010 → L4B3C2_107.

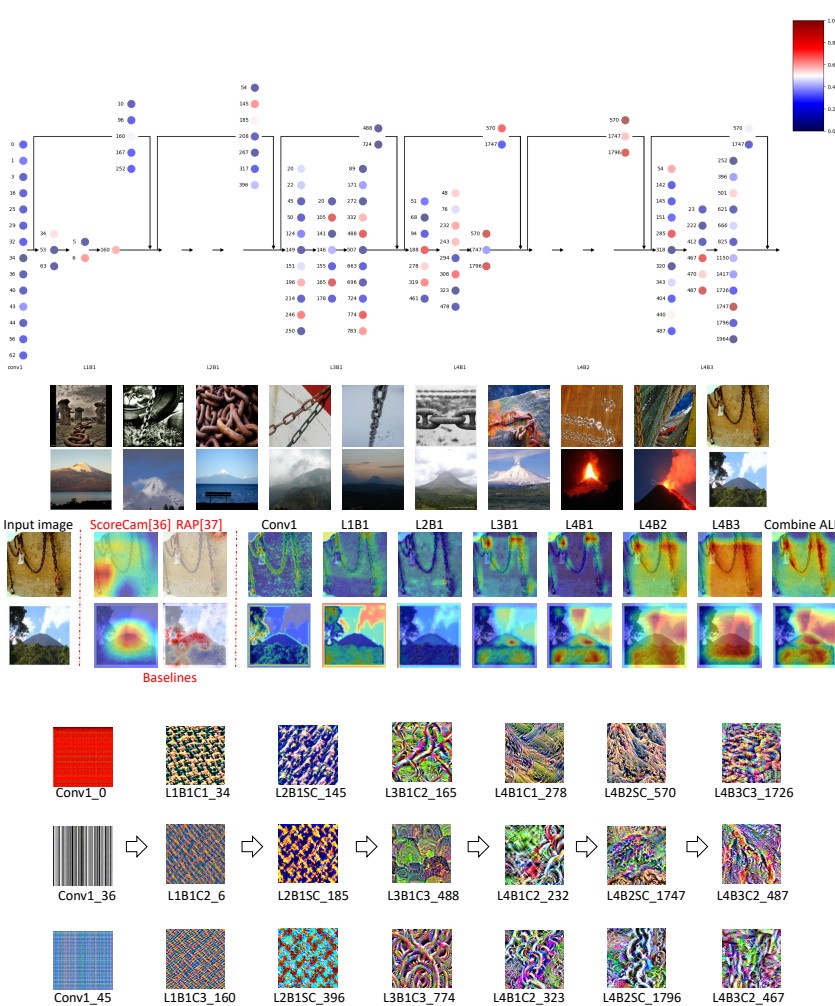

Figure 20: Inference chain by NeuroChains for ResNet-50 when applied to images of "chain" and "volcano". **Top:** The sub-network retains only 18/67 layers and 114/26560 filters. **Middle:** In the SMOE featuremaps, not only the main body of the volcano, but the crater is also highlighted as key features to identify volcanos. **Bottom:** In the first several layers, L3B1C2_165, L3B1C3_774 and L3B1C3_488 extract basic patterns such as curved steel bars and the arc of mountains, whilst deeper layers focus on more global patterns such as different orientations of the folded strata (L4B1C1_278 and L4B2SC_570) and chains (L4B2SC_1796 and L4B3C2_467). L4B3C2_487 captures the features when lava erupts from volcanos as in the penultimate image. It reveals an inference chain for volcano: L2B1SC_145 → L3B1C3_488 → L4B1C1_278 → L4B2SC_570 → L4B3C2_487.

