# OpenReview forum: "Extract Local Inference Chains of Deep Neural Nets"
_ICLR.cc/2021/Conference — Reject_

### Official Review · AnonReviewer3 · 2020-10-23
**Interesting but a rather ad-hoc method, experiments not sufficiently convincing**

**Rating:** 5
**Confidence:** 2

**Review:**

The paper proposes a method called NeuroChains for extracting a sub-network from a deep neural network (DNN) that can accurately match the outputs of the full network for inputs in a small region of the input space. The goal is to be able to explain the important steps that a DNN takes to get from inputs in a small region of the input space, to its predictions for those inputs. NeuroChains initialises the sub-network as the original DNN except each weight/filter is multiplied with a real-valued score. The L1 norm of these scores is minimised while penalising any deviation of the output of the sub-network from the output of the original network. After the minimisation, weights/filters with a score below some threshold value are removed, leaving a sub-network. In the paper there are experiments that aim to verify three claims: (1) NeuroChains can find sub-networks containing less than 5% of the filters in a deep convolutional neural network while preserving its outputs in some small region of the input space; (2) every filter selected by NeuroChains is important for preserving the outputs and removing one of them leads to considerable drop in performance; (3) the sub-networks extracted for small regions of the input space can be generalized to unseen samples in nearby regions.

Strengths:
1. The experimental results suggest that NeuroChains can, as claimed, extract sparse sub-networks that match the outputs of the original network for some small region of the input space. These sub-networks are easier to analyse than the full network, which should help with interpretability.

2. The descriptions of the method and the experiments were very clear. As a result, the experimental results could be reproduced based only on the information in the paper.

3. The distinction between NeuroChains and related methods was made clear. In particular, comparisons were made to pruning methods and work on interpretable machine learning.

Suggestions and questions:
1. In the paper it says that scores are not limited to [0,1] due to possible redundancy among filters in the original DNN. If there are redundant filters, their scores could be set to 0 so that these redundant filters are ignored. Why does the possibility of redundant filters motivate unconstrained scores?

2. The KL divergence between the full network output distribution and the sub-network output distribution was used to penalise the scores. Why was KL divergence used over cross-entropy? The KL divergence is equal to the cross-entropy of the sub-network output distribution relative to the full network output distribution minus the entropy of the full network output distribution. The entropy of the full network output distribution is independent of the scores. Therefore, cross-entropy and KL divergence have the same minima and gradient with respect to the scores. Since the cross-entropy is a little cheaper to compute than the KL divergence, wouldn’t it be better to minimise the cross-entropy instead of the KL divergence?

3. In section 4, paragraph 1, the paper states "(2) every filter selected by NeuroChains is important to preserving the outputs since removing one will leads to considerable drop in performance". However, in Figure 3 (right) there are many filters with scores greater than the threshold tau = 0.1 that appear to cause almost 0 change in the predicted probabilities when removed. Therefore, the claim that all filters found by NeuroChains are important is not well supported by the experimental results.

4. In Figure 3 (right), the magenta line appears to be a line of best fit, which does not by itself imply correlation. However, in the paper it says "magenta line implies strong correlation between the two”. If the goal is to demonstrate strong correlation, wouldn’t it be better to report the Pearson correlation coefficient or Spearman correlation coefficient instead of the line of best fit?

5. On page 8, paragraph 1, the paper references Figure 4 and states that sub-networks extracted for local regions of the input space can generalise to nearby regions because their test fidelity (accuracy with which the output of the full network can be reproduced by the sub-network) remains high when the number of nearest images in the validation set is below 100. However, in my opinion, Figure 4 seems to suggest that as the number of nearest images increases, the test fidelity starts to decrease immediately and at a roughly constant rate until 180 nearest images is reached. Therefore, I don't think that Figure 4 provides strong evidence that the sub-networks extracted for local regions of the input space generalise well to nearby regions.

Overall, the NeuroChains method appears to do a good job at extracting sub-networks from large DNNs and these sub-networks make predictions of the original DNN significantly easier to interpret. However, I don’t fully understand some of the decisions made in the design of the algorithm (questions 1. and 2.) and I am not totally convinced by some of the conclusions drawn from the experimental results (questions 3., 4. and 5.).

---

> ### Author Response · Authors · 2020-11-23
> **Response to AnonReviewer3:**
>
> Thanks for your comments and efforts for reviewing our paper! In the following, we provide answers to your questions.
>
> **1. In the paper it says that scores are not limited to [0,1] due to possible redundancy among filters in the original DNN. If there are redundant filters, their scores could be set to 0 so that these redundant filters are ignored. Why does the possibility of redundant filters motivate unconstrained scores?**
>
> In the original neural net, the redundant filters can contribute important information to the reasoning process. If directly setting their weight to 0, the original neural net’s performance may severely degrade. We call them redundant filters if the information of their featuremaps can be preserved/compressed in one filter’s featuremap, i.e., given this single filter and if we strengthen its weight in the sub-network properly, the redundant filters can be removed. An extreme example is that the redundant filters are all duplicates of a filter: we can preserve only one of them in the sub-network, but its score should be the number of duplicates. In **Figure 6** of the revision, we present two examples of redundant filters and show that a preserved filter combines the functionalities of the redundant ones.
>
> This motivates the unconstrained scores since we need to assign scores > 1 to filters that combine multiple other filters in order to reduce the size of sub-networks. If we limit the score of each filter in (0, 1), these redundant filters will not be pruned (since they contribute important information) and the generated sub-networks will be too large to interpret.
>
> **2. Wouldn’t it be better to minimise the cross-entropy instead of the KL divergence?**
>
> The KL divergence is computed between the output probabilities of the original network and the sub-network. In optimization, it equals the cross-entropy that uses the original network’s output probabilities as the soft labels, up to a constant.
>
> **3. In Figure 2 (right) there are many filters with scores greater than the threshold tau = 0.1 that appear to cause almost 0 change in the predicted probabilities when removed. Therefore, the claim that all filters found by NeuroChains are important is not well supported by the experimental results. Wouldn’t it be better to report the Pearson correlation coefficient or Spearman correlation coefficient instead of the line of best fit?**
>
> In the revision, we updated **Figure 2** based on new experiments on uniform samples. We also generate the same plot for VGG-19 in **Figure 15** of the Appendix. They are clearer in showing the correlations. The slope of the magenta line in the plot is indeed Pearson coefficient, while the shaded area around the line shows the confidence interval. These results provide more evidence of the strong correlation between the score of the removed filter and the resultant change on the output prediction.
>
> **4. Figure 2 seems to suggest that as the number of nearest images increases, the test fidelity starts to decrease immediately and at a roughly constant rate until 180 nearest images are reached. Therefore, I don't think that Figure 3 provides strong evidence that the sub-networks extracted for local regions of the input space generalise well to nearby regions.**
>
> Each sub-network here is extracted based only on 20 samples and **Figure 3** shows that the sub-network can be generalized to 100 unseen samples, on which it can still produce predictions consistent with the original network. Considering the sparsity of data distribution in the high-dimensional space (by curse of dimensionality), the generalization performance is sufficiently good for interpretation. Moreover, note in this setting, **each class only has 40 images**, so the K-nearest neighbor for K>100 with high probability has a very different class label.

---

### Official Review · AnonReviewer2 · 2020-10-29
**interesting set-up and well-written**

**Rating:** 6
**Confidence:** 3

**Review:**

Summary:
The submission considers the task of extracting a small sub-network of a pre-trained neural network that can explain a local region of the data space (a small number of the same classes). The resulting sub-network can be interpreted as the inference path moving from the input to the prediction and the filters and associated weights along this path can be visualised. The key of the proposed approach is to apply a multiplicative weight to each filter and layer in the network and enforce these weights to be sparse. The subnetwork is then selected by thresholding the weights. Some quantitative and qualitative analyses were provided.

Assessment:

The paper considers an interesting interpretation and pruning setting: we wish to prune a pre-trained network to explain how the network performs prediction for a class by training on examples from this class only whilst keeping the pre-trained weights fixed. The resulting final sub-network is very small and thus can be easily visualised. The proposed objective is also very fast to train since the local region used is very small. Despite the similarity to existing post-hoc network pruning/slimming work, I think the goal of finding this small network is very different from that of pruning so thus this work could be of interest to the ICLR community.

The quantitative analysis of the fidelity and the faithfulness of the objective in preserving the output is useful to understand the proposed approach.

The paper is also generally well-written.

Concerns:

+ since the target of the final subnetwork is explainability/visualisation, could this be compared to existing methods in some quantitative form? Could this be used to identify problems and provide more insights on how the network works on out-of-distribution examples or examples with wrong predictions that other methods do not provide? How do we know if the final subnetworks are in general more explainable/interpretable, and what can we do with these sub-networks? (I’m trying to be pedantic here to generate discussion and for me to understand the goal this paper is trying to address)

+ the proposed algorithm has several hyperparameters that seem to have been manually selected (?). How were these values selected and how sensitive is the visualisation/final sub-networks to different hyperparameters?

Overall I think this is a good submission. Adding a comparison to alternatives and discussing more about how this could be used/useful in practice will greatly strengthen the submission.

---

> ### Author Response · Authors · 2020-11-23
> **Response to AnonReviewer2:**
>
> Thanks for your comments and efforts for reviewing our paper! In the following, we provide answers to your questions.
>
> **1. Since the target of the final subnetwork is explainability/visualisation, could this be compared to existing methods in some quantitative form?**
>
> Although our method is for explainability/visualization, our task is different from what the existing interpretation methods targets. Our method’s output is a sub-network, which is hard to compare with existing methods whose output is usually a heatmap. Moreover, the input setting is also different: our method aims to find the inner reasoning process of DNN models on the given local region of data while most previous methods only generate interpretation for a single sample. Hence, in experiments, we can only present the outputs of existing methods in line with our outputs but cannot compare them quantitatively.
>
> We used the metrics/criteria proposed by previous interpretation papers to evaluate NeuroChains. We conducted three quantitative analyses in our paper.
>
> **2. Could this be used to identify problems and provide more insights on how the network works on out-of-distribution examples or examples with wrong predictions that other methods do not provide?**
>
> When applied to examples with wrong predictions, NeuroChains can identify problems and provide explanations of the mistakes made by DNNs. We presented a case study of wrong predictions in **Figure 7** of our revision. NeuroChains is developed to generate interpretations for samples from a local region, otherwise a sufficiently sparse sub-network may not exist, so we are not sure how to apply it to out-of-distribution examples (which by large chance is outside of the local region).
>
> **3. How do we know if the final subnetworks are in general more explainable/interpretable, and what can we do with these sub-networks?**
>
> We can visualize the small sub-networks to reveal the reasoning process of the original network, while directly visualizing each step in the original network is impossible. Our method selects the important connections between filters, which explain how the DNN generates the prediction, step by step, filter by filter, layer by layer. Thereby, we can find which layers are useful and at which filter the inference begins to go wrong. In contrast, previous methods mainly focus on the interpretability of input space which treats the DNN as a black box. The three quantitative analyses (we added more in the revision) provide quantitative evaluation of the sub-networks’ interpretability.
>
> The obtained sub-networks can provide more transparent interpretations to DNNs applied in computer vision, financial services, healthcare services, security services, etc. In these areas, trial and error costs a lot. Using our method, it will help us understand the decisions of the model and avoid risks. Given the interpretations of the reasoning process in DNNs, we can verify them using human expert knowledge and may reversely raise new conjectures for experimental verification. If the interpretations do not make sense, we can improve the DNNs to avoid such mistakes.
>
> **4. How these hyper-parameters were selected and how sensitive is the visualisation/final sub-networks to different hyper-parameters?**
>
> We did not use grid search or other methods to fine-tune these hyperparameters. We only tried a limited number of choices on tens of VGG-19 experiments, and chose the best combination balancing the fidelity and sub-network size, and then applied it to all other experiments without further tuning. In particular, we tried $\tau\in\{0.01, 0.1, 0.5\}$, $\lambda\in\{0.001, 0.005, 0.01, 0.1\}$, and $\lambda_g\in\{1, 2, 5\}$. We finally chose $\tau=0.1, \lambda=0.007, \lambda_g=2$ and this choice performs consistently well and robust on all other experiments.

---

### Official Review · AnonReviewer1 · 2020-10-30
**Nice analysis but with a few gaps**

**Rating:** 6
**Confidence:** 3

**Review:**

This submission extracts a very sparse subnetwork from an RNN. The extraction tries to preserve the computation of the original network in the subnetwork so the behavior of the original network can be analyzed on this subnetwork.

The submission proposes to multiply every filter with a weight/gate and optimize these weights to extract the subnetwork. Up on that, the method also multiple weights to entire network layers so a network layer can also be pruned.

I have a few questions/comments about the proposed methods
1. I understand that it is not easy to optimize binary variables, but can you limit the value of gates in (0, 1) by applying sigmoid transformations of free variables? Is it possible to use some simple rules to prune filters, e.g. remove non-activating entries in max-pooling before running the proposed algorithm?

2. When you remove one layer, do you change behaviors later filters because they actually sees different feature maps in the subnetwork. If this is the case, is the analysis of these filters still applicable to the original network?

3. It seems that gate values of the subnetwork are part of the subnetwork in predictions. "we further fine-tune the nonzero scores" Since feature maps are multiplied to scores, which is not in the original neural network. The multiplication will change feature maps and thus be likely to change behaviors of later filters. Actually, some important filters may not even be selected.

To answer question 2&3, is it possible to compute the correlation between the firing behaviors of filters in the subnetwork and those in the original network? If they are highly correlated, then the claim of that the subnetwork preserves the computation is more consolidated.

4. In figure 3 right, "It shows that the sub-networks suffer from more degeneration if removing a filter with higher score " -- I don't see a strong correlation. The line shows the direction but not the strength of the correlation. The data points are far from this line.

5. I don't see many unexpected findings from this analysis. The visualization of filters and feature maps are somewhat similar to previous analysis. What are new conclusions from the analysis if I omit the points you want to make?

In general, I feel that the research in the direction might be able to provide interesting tools for analyzing CNNs, but there seems to be a few gaps to claim that fidelity of the analysis.

---

> ### Author Response · Authors · 2020-11-23
> **Response to AnonReviewer1:**
>
> Thanks for your comments and efforts for reviewing our paper! In the following, we provide answers to your questions.
>
> **1. Can you limit the value of gates in (0, 1) by applying sigmoid transformations of free variables?**
>
> We considered this option but did not choose it since it cannot combine filters that extract similar or related patterns, while the interpretation in our method needs to reduce the number of filters as much as possible. Two examples have been given in **Figure 6** to compare the featuremaps of the original filters and the combined featuremap produced by the filter with score > 1 in the sub-networks. Another explanation is that limiting the score between [0,1] places more constraint to the optimization problem and the corresponding solution is sub-optimal in terms of the objective.
>
> **2. Is it possible to use some simple rules to prune filters, e.g. remove non-activating entries in max-pooling before running the proposed algorithm?**
>
> In **Figure 12**, we compare the capability of preserving the original neural network’s outputs between NeuroChains and magnitude-based pruning (removing the filters whose output featuremaps’ magnitude (L2 norm) averaged over all considered samples is small). In particular, under the same setting of each experiment in the paper, we prune the original VGG-19 and retain the filters with the largest featuremap magnitude, 180 in total (more than 157(mean)±43(std) filters for sub-networks extracted by NeuroChains), and we then fine-tune the filters’ scores/weights as we did for NeuroChains. Figure 12 shows the histogram of the KL divergence between the original output class distribution and the one produced by the sub-networks. For sub-networks generated by NeuroChains, the KL-divergence in most cases stays close to 0, while the output preserving capability of simple pruning is much worse.
>
> Does this answer your question? We are not sure how to remove non-activating entries in max-pooling to prune the whole filters, so we tried magnitude-based pruning here, which is a simple and widely known pruning rule.
>
> **3. Is it possible to compute the correlation between the firing behaviour of filters in the subnetwork and those in the original network? The behaviour of these filters may be changed after pruning.**
>
> The firing behaviour of the same filter in the original network and the sub-network is usually different since NeuroChains combines similar filters by assigning scores/weights > 1 to some filters. In this way, we can better compress the networks to sub-networks with much fewer filters that are easier to interpret while preserving the original network’s inference process. For this reason, one filter’s featuremap produced in the sub-network is usually a combination of the featuremaps produced by multiple filters in the original network, so the pairwise correlation can be small but cannot reflect this combination behaviour. To see this phenomenon, we present two case studies in **Figure 6**. It shows that each featuremap of the original network’s filter only represents one part of the object, while the featuremap of the preserved filter in the sub-network combines all the parts to form a complete representation of the object in the image.
>
> **4. In figure 2 right, it’s not a strong correlation. The line shows the direction but not the strength of the correlation. The data points are far from this line.**
>
> In **Figure 2** of the revision, we reported the confidence interval to the linear correlation curve and it shows that the linear correlation is strong (the shaded area is small). The fitted linear model and the confidence interval are both automatically generated by seaborn package. However, the linear correlation model is not a perfect model for the joint distribution, which leads to the phenomena you described. Nevertheless, linear correlation is the most commonly used metric and the density contours still indicate a strong correlation between the two quantities.
>
> **5. What are new conclusions from the analysis if I omit the points you want to make?**
>
> 1) NeuroChains produce interpretation for multiple samples from a local region on a data manifold, while most previous methods only generate interpretation for a single sample.
> 2) NeuroChains select the connections between filters, which explain how DNNs generate predictions, step by step, filter by filter, layer by layer. For example, in **Figure 4**, we show an inference chain for Indigo bunting.
> 3) NeuroChains have the capability to select which filters to be interpreted by an interpretation method, and output a score per filter to reflect the importance of each filter;

---

### Official Review · AnonReviewer4 · 2020-10-30
**Interesting idea and missing validations for the regions around a sample and for non-high confidence samples**

**Rating:** 6
**Confidence:** 4

**Review:**

content:
It is about pruning for explanation. The goal of the methods presented is, given a sample x, to extract a network, which is
an unmodified subset of the original network,
and has similar predictions to the original network in a region around x.

The authors derive a gradient-based optimization procedure and do a heuristic threshold-cutoff postprocessing to remove layers and filter channels after the optimization.

The authors evaluate faithfulness in the sample itself.
Faithfulness for the region is evaluated using the nearest neighbors from the dataset and in a high level feature map metric.

strength:
paper concept is well explained. Clarity of the idea.
The outcome are sample-dependent very small sub-networks.



weaknesses:

--The validation of the method.

First of all, they do not validate the impact on the region of a sample x sufficiently, and that must be done because it is a central claim of the paper. The not satisfactorily attendance to that claim is the main reason to reject this paper at the moment. page3: "(3) it is for data from a local region instead of the whole data distribution."

That is relevant even more so as the input space has usually some adversarial samples nearby with respect to a metric in the input space.


Figure 4 gives a partial result - but for a very high level feature space notion of neighbors rather than with respect to a metric in the input space.

Using the last layer for defining the metric to obtain the nearest neighbors may result in rather "semantic" neighbors with similar high level structure, but very different low level structure, which does not conform to the idea of a \eps-lp-metric Ball around x in input space. It is not local with respect to the input space.

In that sense the evaluation of local faithfulness is not complete.

Furthermore using nearest neighbors from the dataset also does not guarantee that they are locally close to x (for regions with low data density this may not be guaranteed)..


--one needs to perform evaluation with some kind of sampling of points within a ball around a sample, close to the local ReLU linearity zone and evaluate faithfulness for those sampled points -- adressing the notion of points being close wrt.~a metric in the input space.

The reviewer would be satisfied, if that would be done for a few hundred test points if 1500 x2 networks costs too much time.

--one needs to perform evaluation on what happens with predicted labels of adversarial samples close to x. At least to consider how likely do they switch back to the original, non-adversarial label when looking at the extracted subnet.

The reviewer would be satisfied, if that would be done for one adversarial per test point but over hundreds of different test points.
Optionally to consider how likely do they switch to another wrong label (this relevant for targeted attacks only, thus optional).

-- any test to be done also for both nets


Secondly,

they do evaluate faithfulness in the sample point. The reviewer is not fully satisfied with the metric.

Fig 3 decrease in probability for predicted class does not tell if it changes the predicted label. They circumvent this problem by using only high confidence samples, however this creates a biased or limited evaluation (namely how this method works on the most confident samples).

it would be better to measure two things:

-- do these evaluations for all samples on their predicted label (a few hundred ...), not only the very confident ones.

--the change in difference to the highest scoring other class ( this gets negative if the predicted label switches ) when comparing original and extracted net.

For example by A quotient of  "diff to highest scoring other class (extracted)" / "diff to highest scoring other class (original)" - this is signed and gets negative if on the extracted net the highest scoring other class is the flipped predicted label

--the probability that the predicted label switches when looking at the subnet



central suggestions for improvement:

run experiments:

--one needs to perform evaluation with random sampling of points within a ball around a sample, close to the ReLU linearity zone and evaluate faithfulness for those sampled points -- adressing the notion of points being close wrt.~a metric in the input space.

--one needs to perform evaluation on what happens with predicted labels of adversarial samples close to x. At least to consider how likely do they switch to the original, non-adversarial label.


Fig3 with:
--the change in difference to the highest scoring other class ( this gets negative if the predicted label switches ) when comparing original and extracted net.

For example by A quotient of  "diff to highest scoring other class (extracted)" / "diff to highest scoring other class (original)" - this is signed and gets negative if on the extracted net the highest scoring other class is the flipped predicted label (as it works on the predicted class on the original net, the denominator is always positive)

--the probability that the predicted label switches when switching at the subnet

-- do not perform the experiments only on high confidence samples but on all samples. This may constitute a bias towards "easy" samples otherwise.

This further also allows to answer the question: are difficult samples with lower confidence less sparsely represented than high confidence samples ?


technical problem:

eq (7) seems to have a typo. The idea is understood, to consider to drop a layer and use the output of the layer before, if input and output size matches, which may make sense for NNs with residual connections.

However if one looks at the case otherwise in eq.7, then it is F^l(x,W^{1:l}) without any filter masks S. In accordance to eq5 this is the mask-less original NN, which is likely a mistake.

It seems that for both cases (incl. the first case) F^l(x,W^{1:l}) needs to be changed to include the masks S
F^l(x,(W^{l'}  \odot S^{l'}, \alpha)^{l'})_{l'=1:l}

That is a fixable problem that does not affect the paper rating.

Technical Questions to the authors:

Section 3.2 algorithms:
-- if the loss is above a certain threshold, it is understood that then you perform the finetuning on the S values. In that case, do you roll back then the filter removal using tau ? In that case, do you roll back then the filter removal by G ?
-- how do you estimate \tau,  \lambda and \lambda_g  ?

General questions to the authors:

The authors contribution in Fig5 and Fig6 seems to be the sparse network extraction, as the SMOE visualization and the filter activation maximization are known. It seems that these visualizations, while nice to show, are not really central to the questions raised by the authors. However, this should not be misunderstood as a wish from the reviewer to remove them. Rather to remark that the paper novelty is the left side in these figures.

The reviewer takes the value of the method so far by: "When applied to samples from a local region in data space, it is plausible that its inference process mainly relies on a small subset of layers/neurons/filters."

--Do the authors have a suggestion how the sparse network extraction can be used for any kind of analysis or insight beyond showing the sparse network itself ?

-- SMOE can also be applied on the original, large network. What is the difference (or value) between applying SMOE visualization on the original network to at first pruning and then applying SMOE on the pruned net?

typos:
--three quantitative analysis over --> three quantitative analyses over (Plural)

--Fig2 is a table

minor suggestions:

--"we remove layer-$\ell$ if G^l < 0.5" is simpler
--remove the top subfig in figure 1, as it is anyway shown in Fig 5 and Fig6 in better resolution. I think you need the space for more interesting content.

Post review: The reviewer thinks that the authors did a thorough job of addressing the reviews. The results are interesting in several aspects, for example Fig 5 and 8. That said, regarding the question "Do the authors have a suggestion how the sparse network extraction can be used for any kind of analysis or insight beyond showing the sparse network itself?" The reviewer has doubts that non-ML expert end users (e.g. M.D.s) could make use of a sparse network as explanation mode (as this would assume that they can make sense of what a neuron has learnt). The reviewer updated his rating upwards.

---

> ### Author Response · Authors · 2020-11-23
> **Response to AnonReviewer4: part 3**
>
> **4. Do the authors have a suggestion how the sparse network extraction can be used for any kind of analysis or insight beyond showing the sparse network itself?**
>
> NeuroChains can be extended to produce domain-specific explanations in addition to the sparse network and can be a bridge in the loop between human and AI. For example, it can be applied to financial services, healthcare, medical diagnosis and security services where neural net models aid the human decisions. Given the interpretations of the reasoning process in DNNs, we can verify them using human expert knowledge and may reversely raise new conjectures for experimental verification. If the interpretations do not make sense, we can improve the DNNs to avoid such mistakes.
>
> Beyond interpretation, NeuroChains can be extended to customize a large-scale neural net for a specific task, e.g., the binary classification tasks shown in our experiments. It can produce light-weight sub-networks preserving the information of the original neural net that is just enough for the sub-tasks and thus can be more easily deployed on edge devices.
>
> These are two examples and there are more possible extensions of NeuroChains which will be studied in our future works.
>
> **5. What is the difference (or value) between applying SMOE visualization on the original network to at first pruning and then applying SMOE on the pruned net?**
>
> In **Figure 11** of the revision, we show two case studies of comparing SMOE generated heatmaps for the original network and the NeuroChains extracted sub-network. We can see that the patterns extracted by the two networks are consistent and are all critical patterns for the class, e.g., the eyes and fists of kangaroos and the feet and face of the horse. However, compared with the original network, these patterns are strengthened in much shallower layers of the sub-network, producing better interpretations. This observation is also consistent with the result of Experiment II.
>
> **Typos and minor suggestions:**
>
> Thanks! We have corrected them in the revision.

---

> ### Author Response · Authors · 2020-11-23
> **Response to AnonReviewer4: part 2**
>
> **Experiment III: Applying NeuroChains to samples with similar ReLU patterns (their ReLU linearity zones are close)**:
> “one needs to perform evaluation with random sampling of points within a ball around a sample, close to the ReLU linearity zone and evaluate faithfulness for those sampled points -- addressing the notion of points being close wrt.~a metric in the input space.”
>
> ReLU pattern does not provide an ideal metric to measure the distance of samples, even in the raw input space, because: (1) the number of ReLU linearity zones grows exponentially with the number of hidden nodes. Most ReLU linearity zones are empty and do not contain any real sample; (2) For the few ReLU linearity zones that do contain samples, each only contains one sample and by large chance its neighboring linearity zones are empty, and this is true for most practical cases as empirical studies suggested. So it is almost impossible to find two samples sharing the same ReLU linearity zone or even close in their ReLU patterns of the first layer; (3) For two ReLU linearity zones that are only different in one facet of their polyhedra (i.e., only one digit of their ReLU patterns flips), their corresponding linear models can still be very different (the linear model is an extreme case of sub-network). Therefore, we speculate that samples close to each other in terms of their ReLU patterns do not share a sufficiently small sub-network preserving their original predictions.
>
> That being said, we evaluated each NeuroChains extracted sub-network in Experiment I on the K-th nearest neighbour (NN) of each sample by sorting the Hamming distance on their ReLU patterns. The K-NN samples’ prediction cannot be well preserved on the sub-networks, because the nearest neighbors in terms of ReLU patterns have very different semantic concepts or classes from the samples that the sub-networks are extracted for. Hence, the local region of ReLU patterns is not a local region on the smooth data manifold. To see this, in **Figure 9** of the revision, for each sample, we computed its L2 distance to the ReLU pattern K-NN sample’s penultimate-layer representation for K=1,2,...10 (the red curve reports mean±std), and we compared them with the L2 distance to the K-NN in the penultimate-layer representation space (the blue curve reports mean±std). It shows that the ReLU pattern K-NN has a much larger L2 distance in the semantic space (i.e., penultimate-layer representation), so it is very different in concept to the original sample. Moreover, we show some examples of the ReLU pattern K-NN images and the penultimate-layer K-NN images for the sample in **Figure 10**, which show that ReLU pattern K-NN images are much less related to the original sample.
>
> **Technical problem:**
>
> A typo in Eq. (7): Thank you for pointing out the typo! The score mask S is lost. We corrected it in the revision.
>
> **General questions to the authors:**
>
> **1. Do you roll back then the filter removal using tau and G after fine-tune?**
>
> No, we don’t roll back to further remove filters after fine-tune. In the fine-tune phase, we fix the preserved filters and the sub-network’s architecture and only optimize the preserved filters’ scores.
>
> **2. How do you estimate \tau, \lambda and \lambda_g?**
>
> We did not use grid search or other methods to fine-tune these hyperparameters. We only tried a limited number of choices on tens of VGG-19 experiments, and chose the best combination balancing the fidelity and sub-network size, and then applied it to all other experiments without further tuning. In particular, we tried $\tau\in\{0.01, 0.1, 0.5\}$, $\lambda\in\{0.001, 0.005, 0.01, 0.1\}$, and $\lambda_g\in\{1, 2, 5\}$. We finally chose $\tau=0.1, \lambda=0.007, \lambda_g=2$ and this choice performs consistently well and robust on all other experiments.
>
> **3. Are difficult samples with lower confidence less sparsely represented than high confidence samples?**
>
> Yes. The number of filters in sub-networks for high-confidence samples is 141(mean)±40(std) while this number for randomly drawn samples in Experiment I is 157(mean)±43(std). Hence, more filters are needed to explain low-confidence samples, but NeuroChains can still produce sufficiently small sub-networks easy to interpret in this case.

---

> ### Author Response · Authors · 2020-11-23
> **Response to AnonReviewer4: part 1**
>
> Thanks for your comments! Please find our new experimental evaluations suggested in your comments and our answers to your questions below.
>
> **Additional experiments required in “weaknesses”:**
> We add 3 groups of experiments according to your comments.
>
> **Experiment I: Applying NeuroChains to the local region  in RAW INPUT space:**
> “one needs to perform evaluation with some kind of sampling of points within a ball around a sample, close to the local ReLU linearity zone and evaluate faithfulness for those sampled points -- addressing the notion of points being close wrt.~a metric in the input space. The reviewer would be satisfied, if that would be done for a few hundred test points if 1500 x2 networks costs too much time.”
>
> As you suggested, we performed 783 experiments each using 20 samples randomly drawn from two classes (not only the high-confidence samples) and achieved 783 new sub-networks by NeuroChains. We evaluated these newly generated sub-networks using the quotient metric suggested by you, i.e., “A quotient of "diff to highest scoring other class (extracted)" / "diff to highest scoring other class (original)” (Eq. (9)). We visualized the result in **Figure 8** of the revision:
>
> The left plot is the histogram of the quotient computed over all the 783 x 20 samples. The histogram shows that most samples keep the original predicted label after pruning, i.e., NeuroChains can preserve the original DNN’s outputs in most cases. Moreover, the number of filters preserved in these sub-networks is 157(mean)±43(std), which is small enough to explain.
>
> The right plot reports the faithfulness of NeuroChains in terms of the quotient’s sign. We remove each filter from each sub-network and report how many samples’ predicted labels are changed after the removal, i.e., the quotient is negative. Each point in the scatter plot corresponds to a sub-network, the x-axis is the score of the removed filter given by NeuroChains, and the y-axis is the proportion of samples with negative quotients. The plot shows a strong linear correlation between the score of the removed filter and the degradation of faithfulness. Since removing filters with high scores results in more samples with predicted class changing after pruning, the score given by NeuroChains measures the importance of filters in DNN inference.
>
> **Experiment II: Applying NeuroChains extracted sub-networks to adversarial examples generated by adversarial attacks:**
> “one needs to perform evaluation on what happens with predicted labels of adversarial samples close to x. At least to consider how likely do they switch to the original, non-adversarial label.”
>
> In order to evaluate NeuroChains on the local regions in the raw input space, we extract sub-networks for random drawn samples and then evaluate the sub-networks on these samples’ adversarial examples generated by two types of attacks: fast gradient sign method (FGSM) and projected gradient descent (PGD). **Figure 5** in the revision compares the robustness of the original neural net (LEFT plot) and the extracted sub-networks (RIGHT plot) under different attacks: each of the plots show the histogram of the output probability for the ground-truth class on those samples (original and adversarial). The left plot shows that the two types of adversarial attack are very effective on the original neural net in reducing the probability of ground truth class. In contrast, the right plot shows that the NeuroChains extracted sub-networks are much more robust to the attacks, because the optimization in NeuroChains not only removes the irrelevant filters but also strengthens the important filters by assigning them weights > 1. This demonstrates the effectiveness of NeuroChains when applied to local regions in the non-smooth raw-input space, and the extracted sub-networks in this case significantly improves the robustness of the original model in defending adversarial attacks.

---

### Author Response · Authors · 2020-11-23
**Summary of changes in the revision**

We appreciate all reviewers for their efforts in reviewing our paper and their constructive comments! We uploaded a revision of the paper that **adds all the experiments required by reviewers and the answers to most questions raised by reviewers**. Here is a summary of the major changes:

1. We add a **new quantitative study about the faithfulness in terms of a new quotient metric** (defined in Eq.(9), as suggested by Reviewer4). We run 783 experiments, each extracting a sub-network for 20 samples uniformly drawn from two classes. The results are reported in Figure 8 of the Appendix. It shows that removing a higher-scored filter changes the predicted class by a larger chance.

2. To study the generalization of the sub-networks in the local region of raw input space, **we evaluate each sub-network on adversarial examples** of the images used to extract the sub-network. The adversarial examples are generated by two types of attacks in the pixel space: fast gradient sign method (FGSM) and projected gradient descent (PGD). We report the result in Figure 5, which shows that the sub-networks are more robust to these attacks than the original network.

3. We discuss **whether the distance between ReLU patterns of two samples is an appropriate metric to define the local regions**. In Figure 10, we show that the K-nearest neighbors in terms of Hamming distance between ReLU patterns can contain very different semantic concepts. Moreover, Figure 9 shows that a sample is usually distant to those ReLU pattern KNNs in semantic space such as the penultimate-layer representation space.

4. In Figure 12 of the Appendix, **we compare NeuroChains with simple magnitude-based pruning** on their capability of preserving the original neural network’s outputs. The result shows that the optimization in NeuroChains cannot be trivially replaced by simple pruning methods.

5. We re-run the experiments to generate Figure 2 and Figure 15 for ResNet-50 and VGG-19 on **uniform samples** instead of high-confidence ones. We add confidence intervals (shaded areas) of the linear (Pearson) correlations (magenta lines) shown in these figures.

6. We discuss that allowing the **scores > 1 is critical** in NeuroChains to generate sufficiently small sub-networks since **it combines the effect of multiple redundant filters** by only preserving one filter with score > 1. This can be illustrated by the new case studies in Figure 6. The figure also explains why the correlation between a filter’s  featuremaps (firing behaviors) in the original network and a sub-network can be small since the sub-network featuremap usually combines multiple pruned filters’ featuremaps.

7. **We compare SMOE heatmaps generated for the original network and the NeuroChains extracted sub-network** in Figure 11 of the Appendix. It shows that the sub-networks can extract important patterns in shallower layers than the original network.

8. We add **a detailed discussion about how the hyper-parameters** in NeuroChains were selected and their sensitivity in our experiments.

---

### Decision · Program_Chairs · 2021-01-07
**Final Decision**

**Decision:**

Reject

**Comment:**

Overall, this seems like a neat idea and well-done work. Main principle is to extract a very sparse net that does a good job at locally "explaining" a given example. The NeuroChains idea does this with a diffentiable sparse objective. I think this work is well-positioned and has nice properties: (1) retains a very small percentage of "filters", (2) it appears that all the selected filters are actually needed/useful (3) there are some generalization properties wrt to unseen samples that are close to the sample of interest.

I appreciate that the authors responded with very detailed rebuttals to the concerns of the reviewers. I'm still worried, like AnonReviewer4, about the generalization around local regions though the follow-up experiments satisfy me for the most part. There is a genuine concern that while this method has the *potential* to produce useful outputs that could be useful for downstream experts to analyze the underlying network, the paper itself doesn't really show this. In other words, while I agree that on the technical side of things, the work passes the bar, it's not clear that the work passes the bar from the impact side of things.

This did make for a genuinely a borderline case in terms of decisions and unfortunately this work landed on the reject side this time around.